# *Saccharomyces cerevisiae* Rev7 promotes non-homologous end-joining by blocking Mre11 nuclease and Rad50's ATPase activities and homologous recombination

**Sugith Badugu[1†], Kshitiza Mohan Dhyani[1†], Manoj Thakur[2], Kalappa Muniyappa[1]\***

[1]Department of Biochemistry, Indian Institute of Science Bangalore, Bengaluru, India; [2]Sri Venkateswara College, University of Delhi, Benito Juarez Marg, New Delhi, India

**\*For correspondence:** kmbc@iisc.ac.in

[†]These authors contributed equally to this work

**Competing interest:** The authors declare that no competing interests exist.

**Sent for Review** 20 February 2024
**Preprint posted** 21 February 2024
**Reviewed preprint posted** 19 April 2024
**Reviewed preprint revised** 06 November 2024
**Version of Record published** 04 December 2024

## eLife Assessment

This manuscript reports **important** data providing evidence that a 42 amino acid region of Rev7 is necessary and sufficient for interaction with the Rad50-Mre11-Xrs2 complex in budding yeast. The authors conclude that Rev7 inhibits the Rad50 ATPase and the Mre11 nuclease with the exception of ssDNA exonuclease activity. The **convincing** data largely support the conclusions, although the effect of Rev7 on homologous recombination is less well documented and the observed effect on resection is moderate. Specifically, the result that the Rev7 C-terminal truncation lacking the 42 amino acid region still suppresses homologous recombination is unexpected and unexplained.

**Abstract** Recent studies have shown that, in human cancer cells, the tetrameric Shieldin complex (comprising REV7, SHLD1, SHLD2, and SHLD3) facilitates non-homologous end-joining (NHEJ) while blocking homologous recombination (HR). Surprisingly, several eukaryotic species lack SHLD1, SHLD2, and SHLD3 orthologs, suggesting that Rev7 may leverage an alternative mechanism to regulate the double-strand break (DSB) repair pathway choice. Exploring this hypothesis, we discovered that *Saccharomyces cerevisiae* Rev7 physically interacts with the Mre11–Rad50–Xrs2 (MRX) subunits, impedes G-quadruplex DNA synergized HU-induced toxicity, and facilitates NHEJ, while antagonizing HR. Notably, we reveal that a 42-amino acid C-terminal fragment of Rev7 binds to the subunits of MRX complex, protects *rev7Δ* cells from G-quadruplex DNA-HU-induced toxicity, and promotes NHEJ by blocking HR. By comparison, the N-terminal HORMA domain, a conserved protein–protein interaction module, was dispensable. We further show that the full-length Rev7 impedes Mre11 nuclease and Rad50's ATPase activities without affecting the latter's ATP-binding ability. Combined, these results provide unanticipated insights into the functional interaction between the MRX subunits and Rev7 and highlight a previously unrecognized mechanism by which Rev7 facilitates DSB repair via NHEJ, and attenuation of HR, by blocking Mre11 nuclease and Rad50's ATPase activities in *S. cerevisiae*.

## Introduction

A hallmark of low fidelity DNA polymerases, also known as DNA translesion synthesis polymerases (TLS polymerases), with no detectable proofreading activity, is their ability to catalyse DNA synthesis across a variety of bulky, helix-distorting DNA lesions (*Prakash et al., 2005*; *Vaisman and Woodgate, 2017*; *Maiorano et al., 2021*; *Ling et al., 2022*). The TLS polymerases are also involved in a plethora of cellular processes, including but not limited to epigenetics, immune signaling, viral mutagenesis,

and cancer development (*Paniagua and Jacobs, 2023*). Indeed, translesion DNA synthesis is a source of mutagenesis, potentially contributing to the development of cancer and drug resistance (*Lange et al., 2011*; *Baranovskiy et al., 2012*; *Pilzecker et al., 2019*). The *Saccharomyces cerevisiae* TLS Pol ζ (henceforth referred to as ScPol ζ ) is a four-subunit enzyme, comprised of catalytic subunit Rev3, two regulatory subunits of Rev7, and the accessory subunits Pol31 and Pol32 (*Johnson et al., 2012*; *Makarova et al., 2012*). Current evidence suggests that the error rate during ScPol ζ -catalyzed replication of undamaged DNA templates is much higher than that of DNA polymerases, as it lacks 3′-to-5′ proofreading exonuclease activity (*Lawrence et al., 1985a*; *Lawrence et al., 1985b*; *Huang et al., 2002*; *Northam et al., 2006*; *Zhong et al., 2006*; *Kochenova et al., 2017*). Consistent with this, *S. cerevisiae rev3*, *rev7*, or *pol32* mutant strains show greatly reduced spontaneous mutation frequencies, driving the notion that ScPol ζ promotes DNA damage-induced mutagenesis (*Quah et al., 1980*; *Lawrence et al., 1985a*; *Lawrence et al., 1985b*; *Morrison et al., 1989*; *Nelson et al., 1996*, *Makarova and Burgers, 2015*).

The TLS Pol ζ exists in a wide range of unicellular and multicellular eukaryotes, including fungi, plants, and animals (*Maiorano et al., 2021*; *Ling et al., 2022*; *Paniagua and Jacobs, 2023*). While the catalytic subunit Rev3 alone is capable of replicating damaged DNA, Rev7 enhances its catalytic efficiency by 20- to 30-fold (*Quah et al., 1980*; *Morrison et al., 1989*; *Nelson et al., 1996*; *Makarova and Burgers, 2015*) and the accessory subunits Pol31 and Pol32 further raise it by 3- to 10-fold (*Johnson et al., 2012*; *Makarova et al., 2012*), suggesting that they abet the processivity of translesion DNA synthesis by Pol ζ (*Acharya et al., 2006*; *Bezalel-Buch et al., 2020*). Indeed, Rev7 (also known as MAD2B and MAD2L2) is an adapter protein, which acts as a bridge between Rev3 and Rev1 (*Haracska et al., 2001*; *Kikuchi et al., 2012*; *Pustovalova et al., 2012*). Various structural and biochemical investigations have uncovered unique structural features of Pol ζ , wherein its subunits interact with each other to form an highly proficient, multi-subunit TLS holoenzyme (*Gómez-Llorente et al., 2013*; *Malik et al., 2020*; *Du Truong et al., 2021*). The high-resolution cryo-EM structures of ScPol ζ holoenzyme have revealed that the subunits Rev3, Rev7, Pol31, and Pol32 assemble into a pentameric ring-shaped structure in which they are maintained by a chain of uninterrupted protein–protein interaction networks (*Gómez-Llorente et al., 2013*; *Malik et al., 2020*; *Du Truong et al., 2021*). Precise details of how Pol ζ holoenzyme achieves its substrate specificity have not been fully understood. Shedding light on this, A. Aggarwal's lab has recently provided insights into the mechanism by which the active site of ScPol ζ responds to the A:C mismatched duplex DNA distortion (*Malik et al., 2022*).

A flurry of research has documented that the tetrameric Shieldin complex – comprising REV7, SHLD1, SHLD2, and SHLD3 – binds single-stranded DNA (ssDNA), blocks 5′ end resection and homologous recombination (HR), antagonizes the recruitment of BRCA1 to the double-strand break (DSB), while facilitating non-homologous end-joining (NHEJ) (*Xu et al., 2015*; *Boersma et al., 2015*; *Mirman et al., 2018*; *Findlay et al., 2018*; *Gupta et al., 2018*; *Ghezraoui et al., 2018*; *Dev et al., 2018*; *Tomida et al., 2018*; *Gao et al., 2018*; *Noordermeer et al., 2018*; *Liang et al., 2020*). Investigations have also shown that the N-terminus of SHLD3 interacts with REV7 using a stereotypical 'safety-belt' interaction mechanism (*Gupta et al., 2018*; *Ghezraoui et al., 2018*; *Dev et al., 2018*; *Tomida et al., 2018*; *Gao et al., 2018*; *Noordermeer et al., 2018*; *Liang et al., 2020*; *Clairmont et al., 2020*; *Dai et al., 2020*). In the alternative pathway, Shieldin-53BP1-RIF1 counteracts DSB resection and subsequent repair by recruiting the CST-Polα-primase complex to promote fill-in at the resected DNA ends (*Mirman et al., 2018*; *Mirman et al., 2022*; *Mirman et al., 2023*). The cells derived from Fanconi anemia patients carrying homozygous mutations in *REV7* display hypersensitivity to DNA cross-linking agents, accumulate chromosome breaks during S/G2 phase and increased p21 levels (*Bluteau et al., 2016*), revealing its critical role in providing protection against FA disease. Some studies have identified non-canonical functions of Rev7/MAD2L2: for example, it binds to and sequesters CDH1, an activator of APC/C, thus prevents premature anaphase onset (*Listovsky and Sale, 2013*; *Vaisman and Woodgate, 2017*; *Ling et al., 2022*). However, a mechanistic understanding of how Rev7 regulates cell cycle events and how such roles differ or relate to its role in TLS remains underexplored. Although the emphasis of the findings differ, the fact that Rev7 functions as an anti-resection factor has spurred a new wave of experiments on DNA repair pathways (*Setiaputra and Durocher, 2019*; *Clairmont and D'Andrea, 2021*).

At DSBs in *S. cerevisiae*, the Mre11–Rad50–Xrs2 (MRX) in conjunction with Sae2 first catalyses endonucleolytic cleavage of 5′-terminated DNA strands and then its 3′→5′ exonucleolytic activity produces a short 3′-ssDNA overhang, which is followed by resection in a 5′→3′ direction by either Exo1 or Dna2–Sgs1 complex to produce long tracks of ssDNA that are critically important for HR (*Cejka and Symington, 2021*). As such, the mechanism by which cells restrain over-resection of DSBs remains incompletely understood, although hyper-resection could potentially hinder optimal HR and trigger genomic instability. Surprisingly, however, Shieldin orthologs are absent in different organisms such as yeast, fruit fly, nematode worm, zebrafish, and frog (*Setiaputra and Durocher, 2019*), raising the crucial question whether the Rev7-mediated regulation of DSB repair pathway choice is evolutionarily conserved in the single-cell eukaryotic organisms such as *S. cerevisiae*. With these observations, we hypothesized that Rev7 in *S. cerevisiae* (hereafter referred to as ScRev7) may recruit an unknown functional equivalent(s) of Shieldin orthologs to regulate the DSB repair pathway choice between HR and NHEJ. Thus, an alternative mechanism might involve the MRX complex on the basis of current knowledge that it plays multiple roles in signaling, processing, and repair of DSBs (*Cejka and Symington, 2021*). In this study, we provide robust evidence that ScRev7, via its 42-amino acid C-terminal fragment in the 'safety belt' region, physically interacts with the Mre11, Rad50, or Xrs2 subunits, protects *rev7Δ* cells from G-quadruplex DNA/HU-induced toxicity and facilitates DSB repair via NHEJ while antagonizing HR. Mechanistic studies revealed that ScRev7 binds to the MRX subunits with sub-micromolar affinity, attenuates Mre11 nuclease and Rad50's ATPase activities, without affecting the ability of the latter to bind ATP. Collectively, our study establishes a previously unrecognized molecular mechanism of regulation of DSB repair, revealing how Rev7 regulates the pathway choice between HR and NHEJ in *S. cerevisiae*.

## Results

### ScRev7 interacts with the MRX subunits

As noted above, studies in cancer cells have shown that the Rev7–Shieldin effector complex facilitates NHEJ by blocking 5′ end resection and HR (*Xu et al., 2015*; *Boersma et al., 2015*; *Mirman et al., 2018*; *Findlay et al., 2018*; *Gupta et al., 2018*; *Ghezraoui et al., 2018*; *Cejka and Symington, 2021*). Since there are no identifiable Shieldin orthologs in *S. cerevisiae* (*Setiaputra and Durocher, 2019*), we began our investigations with a hypothesis that Rev7 may recruit alternative factors such as the MRX subunits to block HR and enable NHEJ. To this end, yeast two-hybrid assay (Y2H) was leveraged for the purpose of studying binary protein–protein interactions between ScRev7 and the subunits of MRX complex, whereas in follow-up studies we mapped the minimal region of ScRev7 required for its association with the MRX subunits. The yeast strain PJ69-4A was co-transformed with plasmids (prey vectors) encoding the Mre11, Rad50, or Xrs2 subunits fused to GAL4 activation domain and ScRev7 fused to the GAL4 DNA-binding domain (bait vector). The positive colonies were selected on SC/-Trp-Leu-His dropout nutrient medium containing 3-aminotriazole (3-AT) (*Fields and Song, 1989*; *James et al., 1996*). Remarkably, we found interactions between Rev7 and the Mre11, Rad50, and Xrs2 subunits (*Figure 1A*), whereas cells bearing empty vector and a plasmid expressing Mre11, Rad50, or Xrs2 subunits did not. Consistent with results from prior research (*Rizzo et al., 2018*), yeast cells transformed with prey and bait vectors expressing Rev7 showed robust growth (*Figure 1A*), indicating the assembly of Rev7 homodimers, which served as a positive control. In an analogous experiment, cells co-transformed with bait and prey vectors expressing Rev7 and Sae2, respectively, failed to grow in different strain backgrounds (bottom panel of *Figure 1A–C*), indicating lack of binary interaction between Sae2 and Rev7. Collectively, these results confirmed the binding specificity of Rev7 to the subunits of MRX complex.

Given that Rev3 has also been implicated in HR-mediated DSB repair (*Sonoda et al., 2003*), we asked whether the MRX subunits interact with ScRev7 in the *rev3Δ* mutant strain. To address this question, the *S. cerevisiae* rev3Δ strain was co-transformed with a combination of bait (pGBKT7 or pGBKT7-*REV7*) and prey vectors (pGADT7, pGADT7-*REV7*, pGADT7-*MRE11*, pGADT7-*RAD50*, pGADT7-*XRS2*, or pGADT7-*SAE2*). Interestingly, we observed binary interactions between the subunits of MRX complex and ScRev7 in the *rev3Δ* mutant in the Y2H system (*Figure 1B*), indicating that Rev3 is dispensable for the binding of ScRev7 to the MRX subunits. Analogously ScRev7 showed

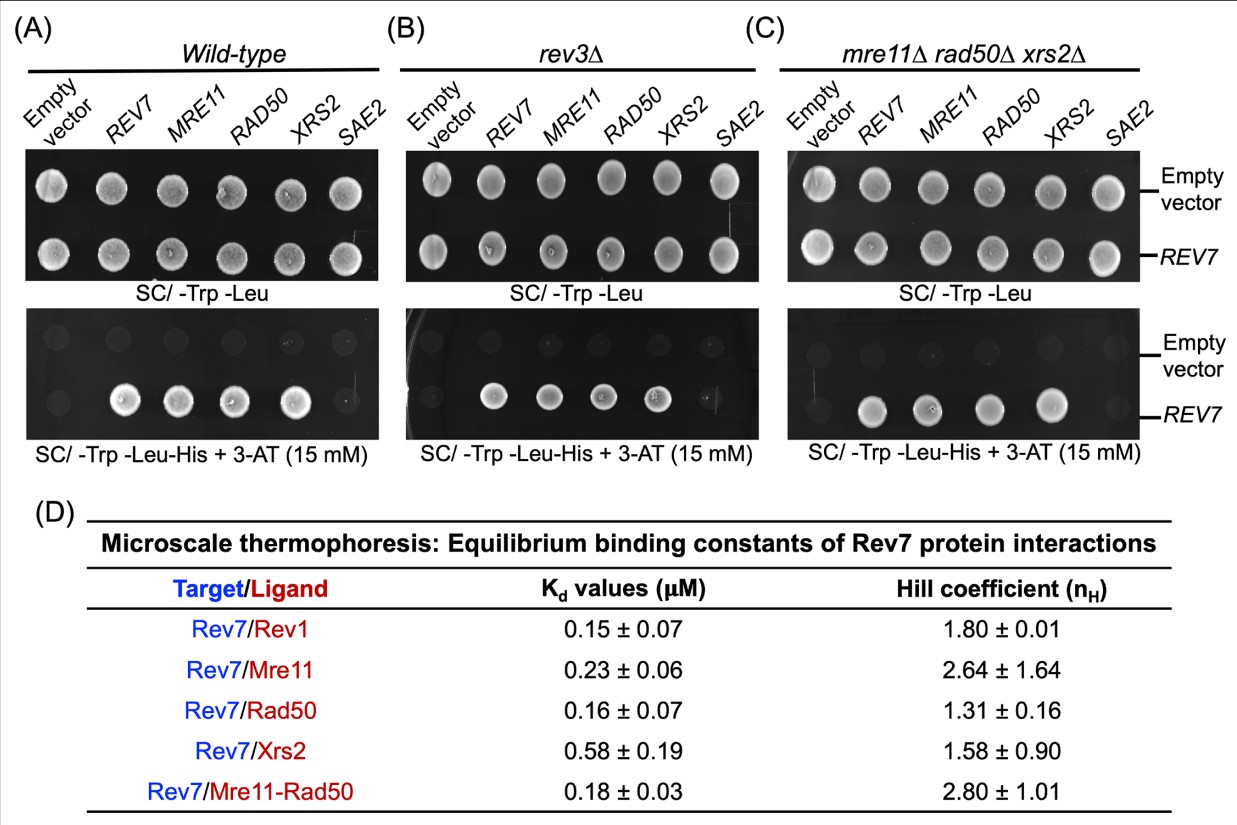

**Figure 1.** Y2H screens suggest interaction between ScRev7 and the MRX subunits. The Y2H assay was performed in (**A**) wild-type, (**B**) *rev3Δ*, and (**C**) *mre11Δ rad50Δ xrs2Δ* mutant strains in PJ69-4A background. These strains were co-transformed with pairwise combinations of empty vector, bait (pGBKT7-*REV7*) and prey (pGADT7/*MRE11*, *RAD50*, *XRS2*, *REV7*, or *SAE2*) plasmids. Equal number of mid-log phase cells was spotted onto the SC/-Trp-Leu agar plates (upper panels) or SC/-Trp -Leu-His agar plates containing 3-aminotriazole (3-AT) (bottom panels). Cells were imaged after 48 hr of growth at 30°C. The images shown in panels (**A–C**) are representative of three independent experiments. (**D**) Quantitative parameters for interaction between ScRev7 and Rev1, Mre11, Rad50, Xrs2, or Mre11–Rad50 proteins.

The online version of this article includes the following source data and figure supplement(s) for figure 1:

**Figure supplement 1.** Sodium Dodecyl Sulfate-Polyacrylamide Gel Electrophoresis (SDS-PAGE) analysis of purified proteins used in this study.

**Figure supplement 1—source data 1.** Original files for gel images and blots displayed in *Figure 1—figure supplement 1*.

**Figure supplement 1—source data 2.** PDF file containing labelled uncropped gel images and blots displayed in *Figure 1—figure supplement 1*.

**Figure supplement 2.** Mass spectromtery analysis of purified Rev7-GFP.

**Figure supplement 3.** Microscale thermophoresis (MST) reveals a direct interaction between Rev7 and MRX subunits.

**Figure supplement 3—source data 1.** Raw data for panels A–F.

**Figure supplement 4.** Rev7-C1 exhibits weak interactions with MRX subunits.

**Figure supplement 4—source data 1.** Raw data for panels A–C.

binary interactions with the subunits of MRX complex in the *mre11Δ rad50Δ xrs2Δ* triple mutant strain (*Figure 1C*), thereby confirming that their association is independent of endogenous MRX subunits.

## ScRev7 physically interacts with the MRX subunits

Microscale thermophoresis (MST) allows for quantitative analysis of protein–protein interactions in free solution (*Wienken et al., 2010*). Since the binding of ScRev7 to the MRX subunits was unanticipated, we sought to validate their interaction by an orthogonal assay and determine their binding affinities using purified proteins. To test our hypothesis, we purified eGFP-tagged ScRev7 (*Figure 1— figure supplement 1A*), confirmed its identity (*Figure 1—figure supplement 2*), and leveraged MST titration approach to measure its binding affinity to purified Mre11, Rad50, Xrs2, and Rev1 subunits, and also to the Mre11–Rad50 complex (*Figure 1—figure supplement 1C, D*). The MST signals were

plotted as a function of ligand concentration (*Figure 1—figure supplement 3A–F*). Normalized MST data were fitted to a logistic binding curve, resulting in an apparent dissociation constants ($K_d$) of 0.16 ± 0.07, 0.23 ± 0.06, and 0.18 ± 0.03 μM for Rad50, Mre11 subunits, and Mre11–Rad50 complex, respectively, which is two- to threefold greater as compared with Xrs2 (*Figure 1D*). *S. cerevisiae* Rev1 was used as a positive control to ensure the accuracy of the Y2H assay. The binding kinetics measured demonstrated that purified Rev1 (*Figure 1—figure supplement 1D*) bound to Rev7 with an affinity (*Figure 1D* and *Figure 1—figure supplement 3D*), comparable to previously reported value (*Rizzo et al., 2018*; *Guo et al., 2003*). On the other hand, negative controls such as purified Sae2 (*Figure 1—figure supplement 1E*) and eGFP showed no significant binding to the GFP-tagged Rev7 and MRX subunits, respectively (*Figure 1—figure supplement 3F*). We next tested the affinity of Rev7-C1 for the Mre11, Rad50, and MR complex. The results showed that Rev7-C1 binds to the Mre11 and Rad50 subunits with about 3- and 8.8-fold reduced affinity, respectively; whereas it binds to the MR complex with ~5.6-fold reduced affinity compared to full-length Rev7 (*Figure 1—figure supplement 4*). Collectively, these results confirm the specificity of interaction between Rev7 and the MRX subunits. It is also noteworthy that the Hill coefficients ($n_H$) indicate larger than one, implying positive cooperativity (*Figure 1D*). The quantitative assessment of binding affinities together with Y2H data suggest that Rev7 robustly interacts with the MRX subunits.

## A 42-amino acid C-terminal segment of Rev7 is critical for its interaction with the MRX subunits

Since the data obtained from the Y2H screening system and MST-based protein–protein interaction assay suggested pairwise association between the subunits of MRX complex and Rev7, we sought to identify the functional domain(s) in the ScRev7 required for interaction with the MRX subunits. For this purpose, we generated three N-terminally truncated and an equal number of C-terminally truncated variants. We refer to these variants as Rev7-N1; Rev7-N2; Rev7-N3, and Rev7-C1, Rev7-C2, and Rev7-C3, respectively (*Figure 2A*). We then asked whether these variants bind to the MRX subunits and enable the growth of yeast cells on selection nutrient medium. Our experiments surprisingly revealed that cells expressing the ScRev7 N-terminally truncated variants showed robust cell growth (bottom panel of *Figure 2B*) similar to the wild-type (WT) (*Figure 1A–C*), indicating that they interact with the subunits of MRX complex. Furthermore, these results indicated that the Rev7's N-terminal HORMA domain (residues 1–149), an evolutionarily conserved protein–protein interaction module, is dispensable for binding to the MRX subunits (*Figure 2B*). Notwithstanding, we do not exclude the possibility that it may play a role that is undetectable by the Y2H assay.

Next, we performed Y2H experiments using the C-terminally truncated species (*Figure 2A*) and found that deletion of C-terminal 42 amino acid residues (i.e., 203–245) resulted in loss of cell proliferation and growth (bottom panel of *Figure 2C*). Similarly, deletion of the C-terminal 150–203 amino acid residues of ScRev7 abrogated yeast cell growth (*Figure 2C*). These results indicate that the C-terminal 42-residue segment of ScRev7 is critical for its interaction with the MRX subunits. To further confirm these results, a Y2H experiment was carried out with cells co-expressing 42 aa peptide and the Mre11, Rad50, or Xrs2 subunits. Such an analysis showed that the ScRev7's 42 residue peptide alone was sufficient for interaction with each subunit of the MRX complex (*Figure 2—figure supplement 1*). However, it remained possible that the inability of cells expressing the C-terminally truncated variants of ScRev7 (with appropriate prey proteins) to grow in selection medium (*Figure 2C*, bottom panel) may be due to altered expression or decreased abundance of truncated species. To explore this possibility, whole-cell lysates derived from cells expressing the N- and C-terminally truncated variants, tagged with c-Myc epitope at the N-terminus, were resolved by SDS–PAGE and probed with anti-c-Myc antibody. Reassuringly, the results revealed comparable levels of N- and C-terminally truncated species of ScRev7 in the whole-cell lysates of strains that were employed for Y2H analyses (*Figure 2—figure supplement 2*).

## Models predicted by AlphaFold-Multimer reveal that Mre11 and Rad50 subunits independently associate with Rev7

To further characterize the interaction between the MRX subunits and ScRev7, a structure prediction algorithm, AF2-multimer (*Evans et al., 2022*), was leveraged to construct structural models of ScRev7–Mre11 and ScRev7–Rad50 heterodimers. The models with the high predicted local distance

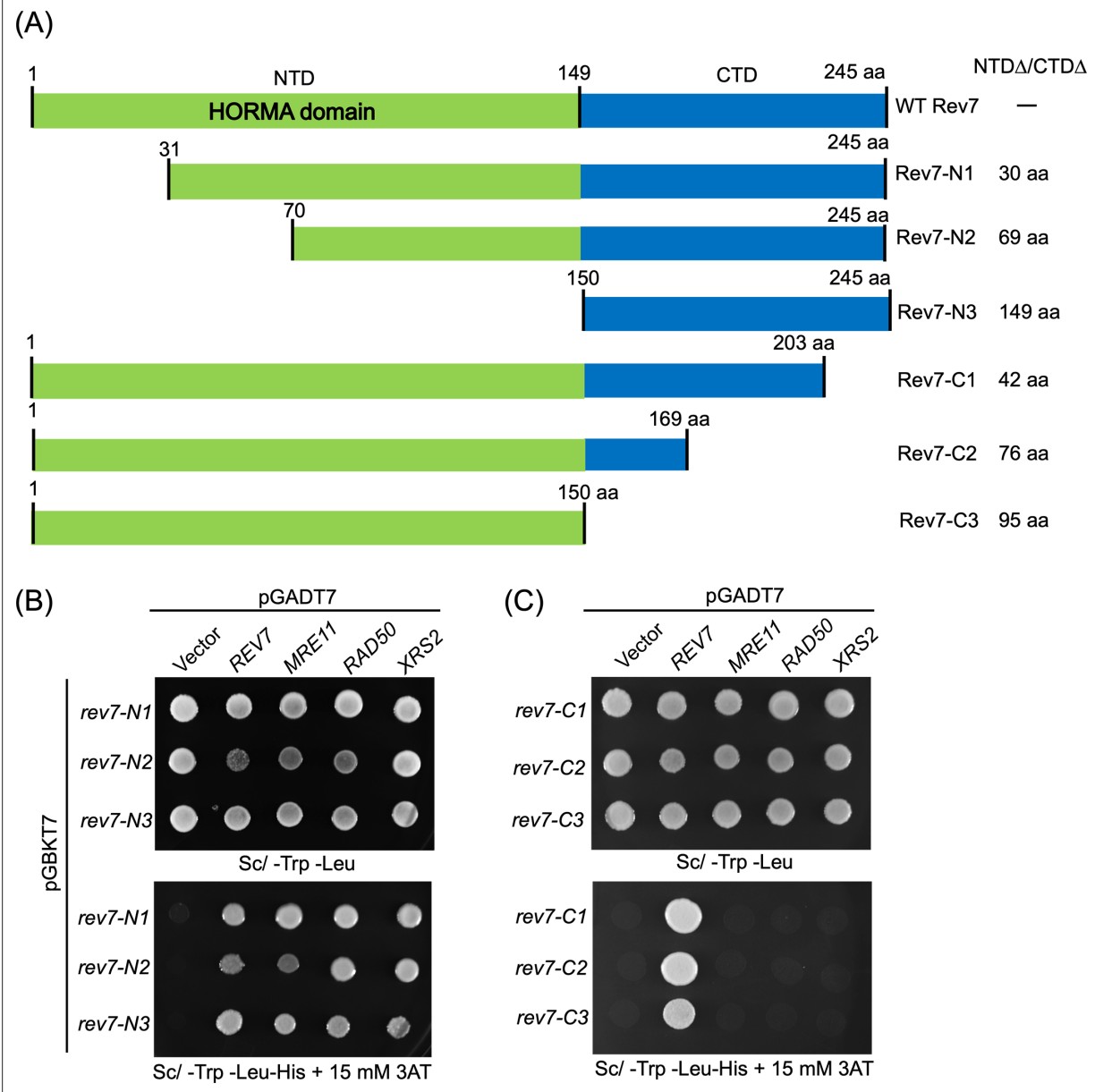

**Figure 2.** Deletion analysis revealed that the C-terminal 42 amino acids of ScRev7 are critical for its binding to the MRX subunits. (**A**) Schematic representation of the full-length and truncated ScRev7 variants. The truncated species lacking the indicated number of amino acids (aa) in the N-terminal domain (NTDΔ) or C-terminal domain (CTDΔ) is indicated on the right-hand side of the figure. (**B**) Representative images of spot assays of cells carrying pairwise combination of empty vector, bait and prey plasmids expressing N-terminally truncated species of Rev7 and full-length Rev7, Mre11, Rad50, and Xrs2, respectively. (**C**) Same as panel (**B**), but with the bait plasmids encoding C-terminally truncated Rev7 variants. Cells were imaged after 48 hr of growth at 30°C. Y2H assay was performed as described in the legend to *Figure 1*. Data are representative of three independent experiments.

The online version of this article includes the following source data and figure supplement(s) for figure 2:

**Figure supplement 1.** The C-terminal 42 amino-acid region of Rev7 interacts with M/R/X subunits.

**Figure supplement 2.** Western blot showing the abundance of N- and C-terminally truncated variants of Rev7.

**Figure supplement 2—source data 1.** Original raw files for western blot analysis displayed in *Figure 2—figure supplement 2*.

**Figure supplement 2—source data 2.** PDF files containing uncropped labeled western blots displayed in *Figure 2—figure supplement 2*.

**Figure supplement 3.** AlphaFold-Multimer generated models of Rev7–Mre11 and Rev7–Rad50 protein complexes.

difference test (pLDDT) scores were considered for further analysis. Strikingly, the models indicated that Mre11- and Rad50-binding surfaces overlap with ScRev7, suggesting that the latter can bind these subunits (*Figure 2—figure supplement 3*). In the model of Rev7–Mre11 complex, residues Asp 206 and Ile 240 in the Rev7 C-terminal 'safety belt region' (green) mediate dimerization with Mre11 through residues Asp 131 and Arg 181. Interestingly, His 127 in the Rev7 N-terminal HORMA domain also contribute its binding to Asp 131 of Mre11 (*Figure 2—figure supplement 3A*, *Supplementary file 1a*). Furthermore, modeling studies of Rev7–Rad50 complex showed that the ScRev7 C-terminal residues Lys 168, Glu 184, Asn 189, and Asp 188 mediate dimerization with ScRad50 via residues Glu 577, Lys 596, and Arg 603 (*Figure 2—figure supplement 3B*, *Supplementary file 1b*). Curiously, the AF2-multimer models also revealed that amino acid residues outside of the 42-residue fragment also contribute to pairwise interactions between Rev7 and the Mre11 and Rad50 subunits, although Y2H assays did not identify such interaction. Indeed, similar findings have been previously noted for several other interacting partners (*You et al., 2006*; *Koegl and Uetz, 2007*; *Hoff et al., 2010*).

## The Rev7's C-terminal 42-residue fragment mitigates the G-quadruplex-HU-induced toxic effects

Several studies have documented the genome-wide prevalence of G-quadruplex DNA structures in various organisms ranging from viruses to humans, which play regulatory roles in diverse cellular processes (*Rhodes and Lipps, 2015*; *Spiegel et al., 2020*; *Yadav et al., 2021*). For instance, it has been shown that Rev1-deficient chicken DT40 cells exhibit defects in replicating G-quadruplex-forming motifs (*Sarkies et al., 2010*; *Sarkies et al., 2012*). Similarly, computational and genetic studies in *S. cerevisiae* have revealed that G-quadruplex-forming motifs cause slow growth in replication stressed Pif1-deficient cells and affect genome integrity (*Capra et al., 2010*; *Paeschke et al., 2011*; *Paeschke*

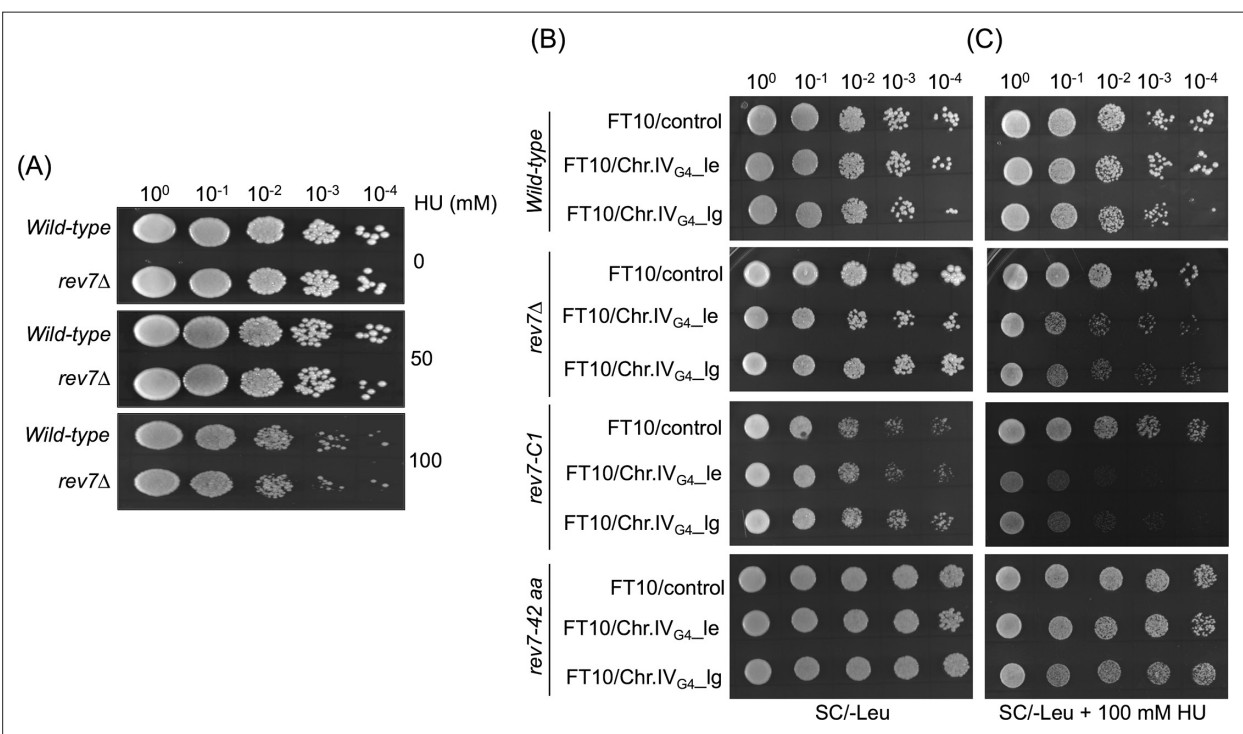

**Figure 3.** A 42-amino acid segment at the extreme C-terminus of ScRev7 renders cells resistant to the synergistic adverse effect of G-quadruplex DNA and HU. (**A**) *REV7* deletion does not affect HU sensitivity of *rev7Δ* cells. Representative images of YPD plates showing spot assay of wild-type (WT) W1588-4C strain and its derivative *rev7* mutant cells, in the absence or presence of 50 or 100 mM HU. (**B**) Representative images of SC/-Leu agar plates showing spot assay of wild-type, *rev7Δ*, *rev7*-C1, or *rev7*-42 aa cells harboring the indicated plasmids carrying G-quadruplex-forming motifs derived from chromosome IV (Chr.IV$_{G4}$). The cells were grown on SC/-Leu agar plates in the absence of HU. (**C**) Same as panel (**B**), but the growth medium contained 100 mM HU. For serial dilutions, each strain was grown in SC/-Leu medium, normalized to OD$_{600}$ = 1.0, and serially diluted using yeast nitrogen base medium. Five μl aliquots from the serial dilutions were spotted onto the SC/-Leu agar plates with or without HU. FT10/Control plasmid lacks the G-quadruplex DNA insert. The abbreviations 'le' and 'lg' stand for leading and lagging strands, respectively. Cells were imaged after 4 days of growth at 30°C. Data are representative of three independent experiments.

*et al., 2013*). Inspired by these findings, we sought to understand whether ScRev7 plays a role in genome maintenance using the assay developed by the V. Zakian's lab (*Paeschke et al., 2011*). We first compared the viability of *rev7Δ* mutant cells devoid of exogenous G-quadruplex-forming motifs with that of WT under conditions of optimal growth and HU-induced replication stress. This analysis revealed that, like the WT, *rev7Δ* mutant cells grew robustly in the absence or presence of HU (*Figure 3A*).

Next, we next sought to determine whether the G-quadruplex DNA motifs affect cell viability in the absence and presence of HU. To address this question, we tested the effect of exogenous G-quadruplex-forming motifs, positioned on either the leading or lagging template strands, on the viability of WT and *rev7Δ* mutant strains, relative to cells expressing the ScRev7 variants (Rev7-C1 or 42-residue C-terminal peptide) on SC/-Leu medium lacking HU. We found that all strains used in this experiment showed comparable growth phenotypes on this medium (*Figure 3B*). On the other hand, *rev7Δ* mutant and the strains expressing the variant Rev7-C1 were highly sensitive to exposure to HU when compared to the WT cells (*Figure 3C*). Remarkably, however, cells expressing the Rev7's 42-residue C-terminal peptide exhibited robust growth in the same medium containing HU (*Figure 3C*). Combining the results presented, we suggest that the 42-residue C-terminal peptide, but

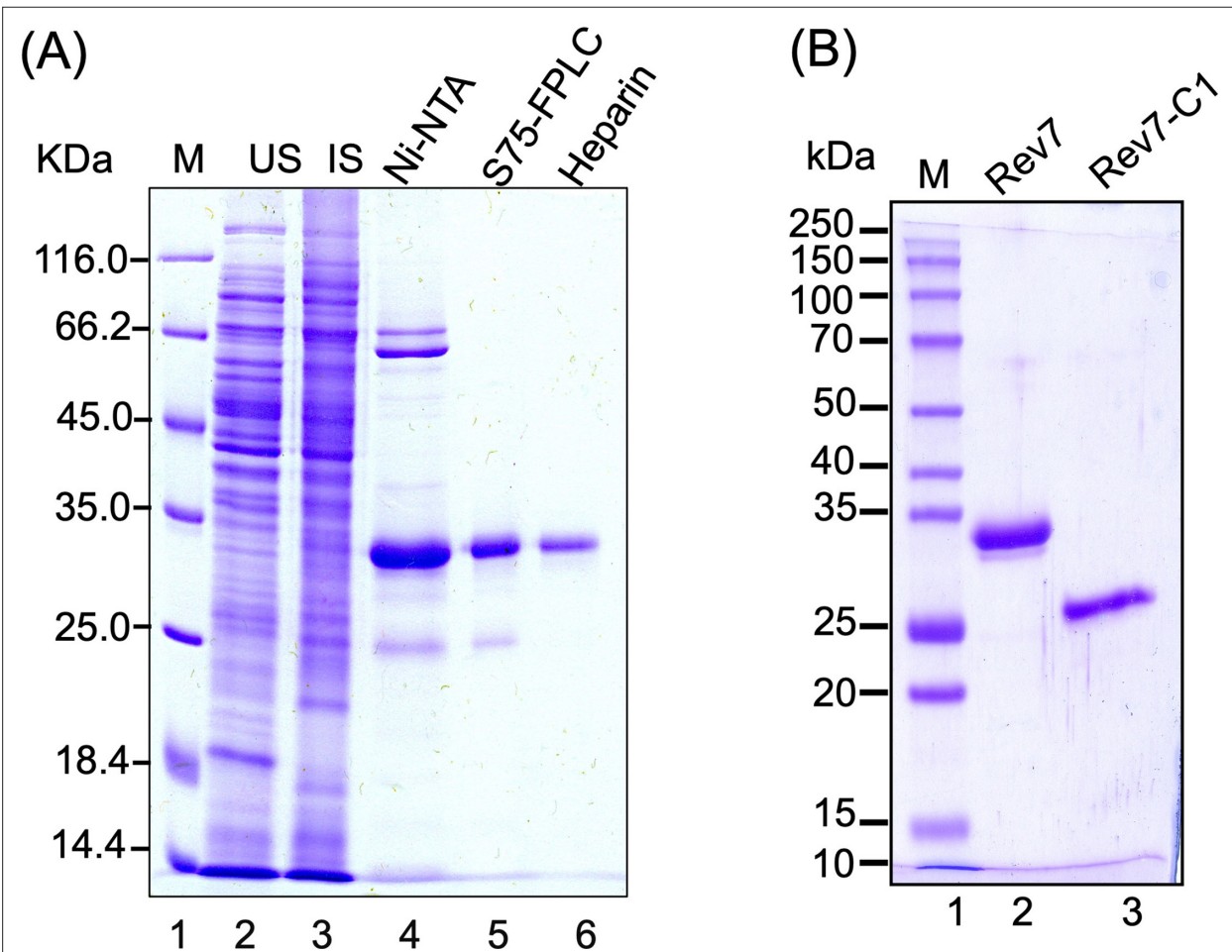

**Figure 4.** Purification of ScRev7 and ScRev7-C1 proteins. (**A**) SDS–PAGE analysis of protein samples from different stages of ScRev7 purification. Lane 1, standard protein markers; lane 2, uninduced cell lysate (10 µg); lane 3, induced cell lysate (10 µg); lane 4, Ni²⁺-NTA column eluate (3 µg); lane 5, Superdex S75 column eluate (1 µg); lane 6, eluate from heparin column (0.8 µg). (**B**) SDS–PAGE analysis of purified truncated ScRev7-C1 variant. Lane 1: standard protein markers. Lanes 2 and 3, purified full-length ScRev7 and ScRev7-C1 variant.

The online version of this article includes the following source data for figure 4:

**Source data 1.** Original files for gel images displayed in *Figure 4*.

**Source data 2.** PDF file containing labelled uncropped gel images displayed in *Figure 4*.

not Rev7-C1 variant, confers protection to cells from the toxic effects of HU/G-quadruplex-forming motifs, regardless of whether they occur in the leading or lagging template strands.

## Rev7 inhibits Mre11 endo- and exonucleolytic activities

Decades of work has documented that Mre11 is a $Mn^{2+}$-dependent bifunctional enzyme with both endo- and exonuclease activities, which are critical for DNA end resection (*Paull and Gellert, 1998*; *Tsubouchi and Ogawa, 1998*; *Ghosal and Muniyappa, 2007*; *Stracker and Petrini, 2011*; *Ghodke and Muniyappa, 2013*; *Paull, 2018*; *Casari et al., 2019*). Furthermore, it has been demonstrated that Mre11 cleaves non-B DNA structures such as DNA hairpins, as well as intra- and intermolecular G-quadruplex structures (*Trujillo and Sung, 2001*; *Lobachev et al., 2002*; *Ghosal and Muniyappa, 2005*; *Ghosal and Muniyappa, 2007*). To further explore the functional significance of complex formation between Rev7 and Mre11, ScRev7 and its truncated derivative ScRev7-C1 were expressed in and purified from *Escherichia coli* whole-cell lysates to homogeneity (*Figure 4A, B*). We then performed an experiment in which equimolar amounts of MRX subunits were incubated

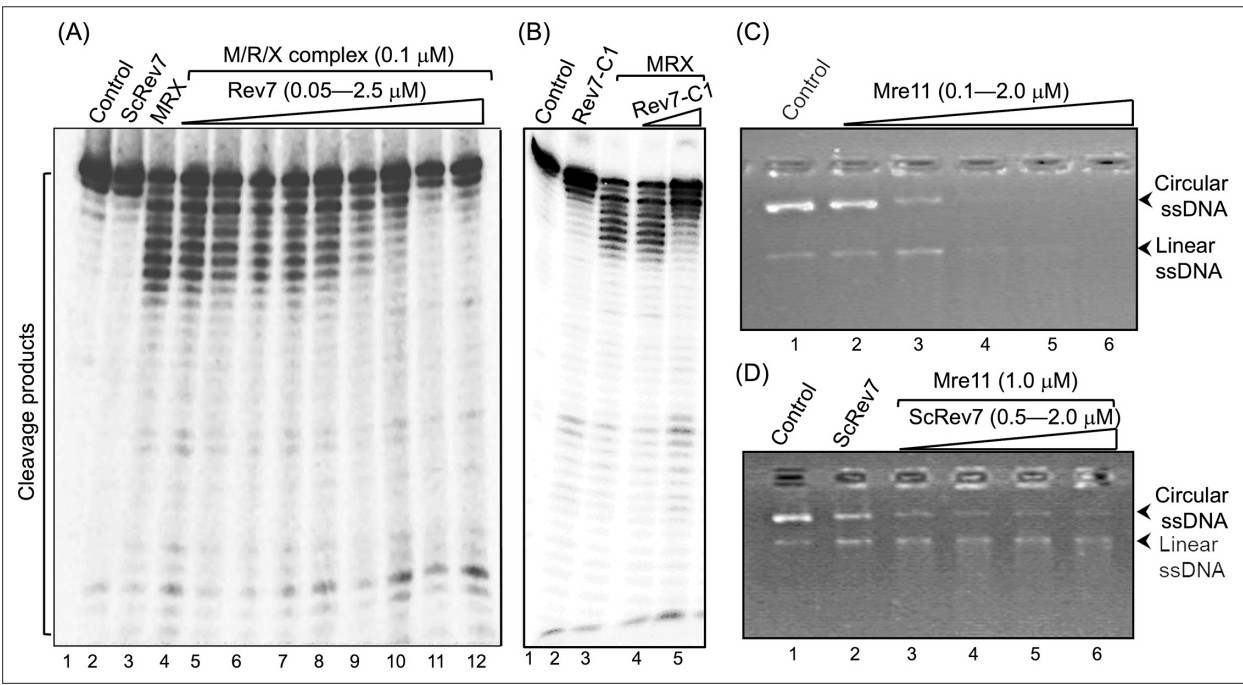

**Figure 5.** ScRev7 impedes both exo- and endonucleolytic activities of the MRX subunits. (**A**) A representative image showing the effect of ScRev7 on the exonuclease activity of MRX complex. Reaction mixtures containing 5nM $^{32}$P-labeled 60bp dsDNA were incubated in the absence or presence of M/R/X subunits and various amounts of ScRev7. Lane 1, $^{32}$P-labeled dsDNA. Lane 2, same as in lane 1, but with 2.5 µM ScRev7. Lane 3, same as in lane 1, but with M/R/X subunits (0.1 µM each). Lanes 4–12, same as in lane 3, but with 0.05, 0.1, 0.3, 0.5, 0.7, 1, 1.5, 2, and 2.5 µM of ScRev7, respectively. (**B**) A representative image showing the effect of Rev7-C1 variant on the exonuclease activity of the MRX complex. Assay was performed as in panel (**A**), but with ScRev7-C1 variant. Lane 1, $^{32}$P-labeled dsDNA. Lane 2, same as in lane 1, but with 2.5 µM ScRev7-C1 variant. Lane 3, same as in lane 1, but with M/R/X subunits (0.1 µM each). Lanes 4 and 5, same as in lane 3, but with 1.5 and 3.0 µM of ScRev7-C1 variant, respectively. (**C**) A representative image showing Mre11 endonuclease activity on circular ssDNA. Reaction mixtures containing 100 ng M13 circular ssDNA were incubated without (lane 1) or with 0.1, 0.5, 1, 1.5, and 2 µM Mre11, respectively (lanes 2–6). (**D**) A representative image showing the effect of ScRev7 on Mre11 endonuclease activity by ScRev7. Reaction mixtures containing 1 µM Mre11 were pre-incubated with 0.5, 1.0, 1.5, and 2 µM of ScRev7 (lanes 3–6, respectively), prior to the addition of 100 ng of M13 circular ssDNA. Lane 1, circular ssDNA alone. Lane 2, same as in lane 1, but with 2 µM of ScRev7. Increasing concentrations of the indicated protein is represented by open triangles at the top of the gel.

The online version of this article includes the following source data and figure supplement(s) for figure 5:

**Source data 1.** Original files of gels displayed in *Figure 5*, panels A–D.

**Source data 2.** PDF file containing uncropped labeled gel images displayed in *Figure 5*.

**Figure supplement 1.** Rev7 does not impact the endonuclease activity of Sae2.

**Figure supplement 1—source data 1.** Original files for gels displayed in panels A and B in *Figure 5—figure supplement 1*.

**Figure supplement 1—source data 2.** PDF file containing uncropped labeled gel images displayed in *Figure 5—figure supplement 1*.

with 5'-$^{32}$P-labeled dsDNA in the presence or absence of ScRev7. The reaction products were analyzed as previously described (*Ghosal and Muniyappa, 2005*). The results informed that (1) ScRev7 has no nuclease activity (*Figure 5A*, lane 2) and (2) Mre11 cleaved $^{32}$P-labeled dsDNA, generating a pattern of DNA fragments in a ladder-like pattern (*Figure 5A*, lane 3). Subsequently, we investigated the effect of ScRev7 on MRX nuclease activity by co-incubating various concentrations of ScRev7 and a fixed amount of MRX complex, prior to the addition of 5'-$^{32}$P-labeled dsDNA. We detected a notable and substantial inhibition in the MRX complex-mediated DNA cleavage activity, and almost complete inhibition at 2 μM ScRev7 (*Figure 5A*, lanes 4–12). We carried out additional experiments to test the effect Rev7-C1 on Mre11 nuclease activity. By comparison, the efficiency of inhibition was expectedly less than that of full-length ScRev7. Curiously, we note that the ScRev7-C1 does not faithfully recapitulate the Y2H results. One possible reason is that amino acid residues in the ScRev7-C1 fragment interact with Mre11 and cause partial inhibition, in good agreement with the AF2 modeling data (*Figure 1—figure supplement 4*, *Figure 2—figure supplement 3*).

Many studies have demonstrated that Mre11 exhibits both endo- and exonuclease activities independently of Rad50 and Xrs2 subunits (*Cejka and Symington, 2021*). Thus, we next examined the effect of ScRev7 on the Mre11 ssDNA-specific endonuclease activity in reactions lacking Rad50 and Xrs2. In accord with previous studies, Mre11 digested all the input ssDNA substrate into small fragments/nucleotides in a manner dependent on its concentration (*Figure 5C*). Interestingly, we found that the addition of increasing concentrations of ScRev7 coincided with a concomitant decrease in the Mre11 endonuclease activity (*Figure 5D*). To ascertain the specificity, the effect of ScRev7 on Sae2's endonuclease activity was investigated. As expected (*Lengsfeld et al., 2007*; *Ghodke and Muniyappa, 2016*), Sae2 exhibited concentration-dependent nuclease activity on dsDNA (*Figure 5—figure supplement 1A, B*). However, ScRev7 did not inhibit Sae2's nuclease activity, even at concentrations by 10 times higher (*Figure 5—figure supplement 1C, D*). Together, these results support the idea that inhibition of Mre11 endonuclease activity is due to its direct interaction with ScRev7.

## ScRev7 impedes Rad50's ATPase activity without affecting its ATP-binding ability

Several studies have demonstrated that the ATPase activity of Rad50 plays an important regulatory role in DNA recombination and repair (*Cejka and Symington, 2021*). Given that ScRev7 specifically associated with Rad50, we asked whether such association affects the ability of Rad50 to bind ATP and catalyze its hydrolysis. To test this possibility, different concentrations of purified Rad50 were incubated with a fixed amount of [γ-$^{32}$P]ATP, and then the reaction mixtures were UV irradiated prior to subjecting the samples to SDS-PAGE analysis. The results showed a single band that migrated as a 153-kDa species corresponding to the position of ScRad50 (*Figure 6A*). Quantitative analyses indicated that Rad50 binds [γ-$^{32}$P]ATP in a manner dependent on its concentration (*Figure 6C*). The results from an accompanying experiment revealed comparable levels of [γ-$^{32}$P]ATP binding by Rad50 in the presence or absence of ScRev7 (*Figure 6B, D*), suggesting that it does not impair the ATP-binding ability of Rad50.

We next explored the potential effect of ScRev7 on ATP hydrolysis catalyzed by Rad50 using thin layer chromatography. Consistent with previous studies (*Ghosal and Muniyappa, 2007*), Rad50 catalyzed [γ-$^{32}$P]ATP hydrolysis to ADP and $^{32}$Pi in a manner dependent on its concentration in the absence of ScRev7 (*Figure 6E, F*). Interestingly, while ScRev7 itself has no ATPase activity, its addition led to inhibition of ATP hydrolysis catalyzed by Rad50 in a dose-dependent manner (*Figure 6G, J*). The results of a parallel experiment indicated that Rev7 does not affect the ATPase activity of a meiosis-specific *S. cerevisiae* Dmc1, even at concentrations by four times higher, indicating its specificity for Rad50 (*Figure 6H, J*). Similar analysis showed that ScRev7-C1 inhibited the ATP hydrolysis of Rad50 but threefold less efficiently than its full-length counterpart (*Figure 6I, J*). These results were validated using a colorimetric molybdate/malachite green-based assay (*Lanzetta et al., 1979*). While ScRev7 inhibited the ATPase activity of Rad50 to an extent of 60% at the highest concentration tested, Rev7-C1 was about threefold less inhibitory than its full-length counterpart at an identical concentration (*Figure 6—figure supplement 1*). Collectively, our results support a model in which ScRev7 negatively regulates the catalytic activities of Mre11 and Rad50 subunits.

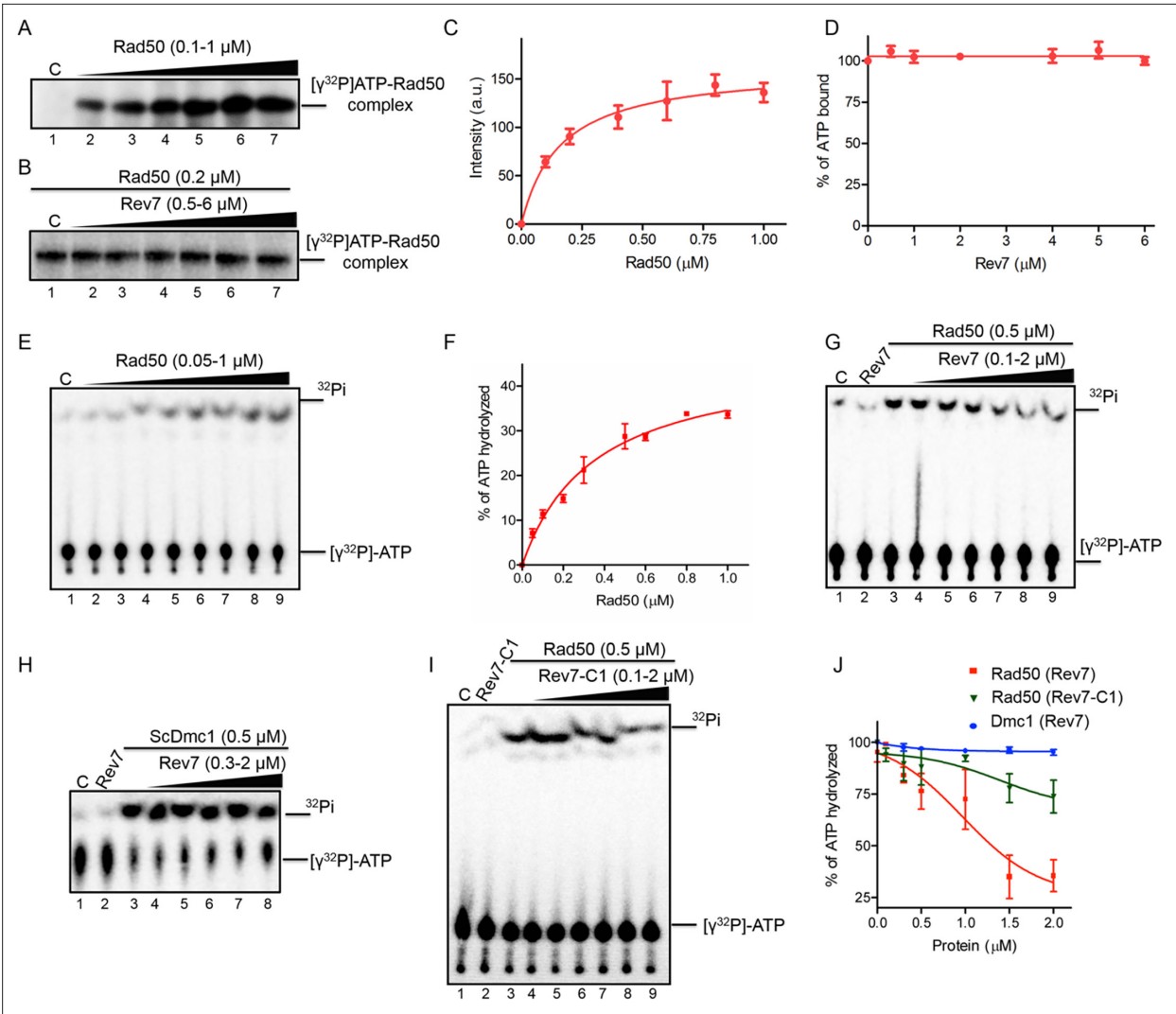

**Figure 6.** Rev7 inhibits the ATPase activity of Rad50 without impacting its ability to bind ATP. (**A**) Rad50 binds [γ-$^{32}$P]ATP in a dose-dependent manner. Lane 1, no protein. Lanes 2–7, reactions were performed with 0.1, 0.2, 0.4, 0.6, 0.8, and 1 µM of Rad50 and 400 pmol[γ-$^{32}$P]ATP. (**B**) ScRev7 does not affect the ability of Rad50 to bind ATP. Lane 1, Rad50 and 400 pmol[γ-$^{32}$P]ATP. Lanes 2–7, same as in lane 1, but with 0.5, 1, 2, 4, 5, and 6 µM of ScRev7, respectively. (**C**) Quantification of ATP binding by Rad50 as a function of its concentration. (**D**) Quantification of the effect of ScRev7 on ATP binding by Rad50. (**E**) ATPase activity of Rad50 as a function of its concentration. Reactions were performed in the absence (lane 1) or presence (lanes 2–9) of 0.05, 0.1, 0.2, 0.3, 0.5, 0.6, 0.8, and 1 µM of Rad50, respectively. (**F**) Quantification of Rad50 ATPase activity as a function of its concentration. (**G**) ScRev7 abrogates ATP hydrolysis catalyzed by Rad50. (**H**) ScRev7 does not impact the ability of ScDmc1 to hydrolyze ATP. Lane 1 contained 400 pmol [γ-$^{32}$P] ATP; lane 2, same as in lane 1, but with 2 µM of ScRev7; lane 3, as in lane 1, but with 0.5 µM of ScDmc1; lanes 4–8, as in lane 3, but with 0.3, 0.5, 1, 1.5, and 2 µM ScRev7, respectively. (**I**) ScRev7-C1 variant impedes ATP hydrolysis catalyzed by Rad50. In panels (**G**) and (**I**), lane 1 contained 400 pmol [γ-$^{32}$P] ATP; lane 2, same as in lane 1, but 2 µM ScRev7/ScRev7-C1 variant; lane 3, as in lane 1, but with 0.5 µM Rad50; lanes 4–9, as in lane 3, but with 0.1, 0.3, 0.5, 1, 1.5, and 2 µM ScRev7/ScRev7-C1 variant, respectively. (**J**) Quantification of the inhibitory effect of ScRev7/ScRev7-C1 on ATP hydrolysis catalyzed by Rad50 or ScDmc1. The closed triangles on the top of gel images in panels (**A**), (**B**), (**E**), (**G**), (**H**), and (**I**) represent increasing concentrations of Rad50, ScRev7, or ScRev7-C1. Error bars indicate SEM, and data are representative of three independent experiments.

The online version of this article includes the following source data and figure supplement(s) for figure 6:

**Source data 1.** Original files of gel images displayed in *Figure 6*, panels A and B.

**Source data 2.** PDF file representing labeled uncropped gel images corresponding to *Figure 6*, panels A and B.

**Figure supplement 1.** ScRev7 inhibits the ATPase activity of ScRad50.

## *REV7* facilitates NHEJ in *S. cerevisiae*

In *S. cerevisiae*, the heterotrimeric MRX complex has been implicated in both Ku-dependent NHEJ and microhomology-mediated end-joining repair (*Moore and Haber, 1996*; *Boulton and Jackson, 1998*; *Ma et al., 2003*; *Zhang and Paull, 2005*). Multiple studies in cancers have shown that Rev7 inhibits DNA end-resection and favors NHEJ over HR (*Gupta et al., 2018*; *Ghezraoui et al., 2018*; *Dev et al., 2018*; *Gao et al., 2018*; *Liang et al., 2020*). To our knowledge, it is unknown whether these findings are relevant to other species. However, we note that *Schizosaccharomyces pombe* Rev7 has been shown to inhibit long-range resection at DSBs (*Leland et al., 2018*). Furthermore, it is unclear whether the *S. cerevisiae* Rev1, Rev3, and Rev7 subunits are required for NHEJ. To investigate this,

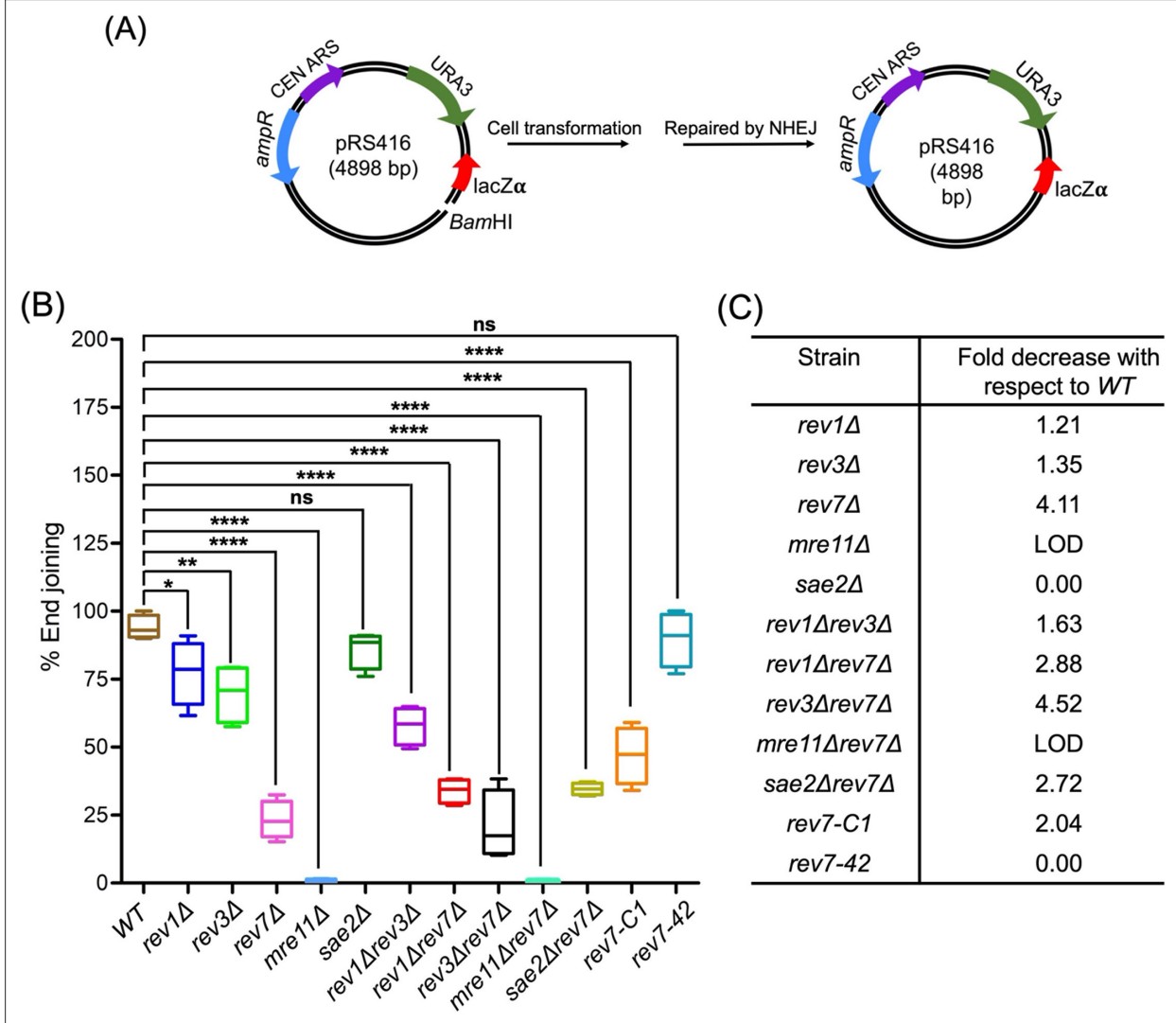

**Figure 7.** *S. cerevisiae REV7* promotes non-homologous end-joining (NHEJ)-mediated double-strand break (DSB) repair. (**A**) Map of the NHEJ reporter plasmid and the experimental workflow. Sixty ng of uncut or linearized pRS416 plasmid DNA was transformed into the wild-type (WT) (W1588-4C) and indicated isogenic mutants carrying single or double deletions. Transformants were selected on SC medium lacking uracil. (**B**) Quantification of NHEJ efficiency relative to WT cells. (**C**) Quantification of fold decrease in the efficiency of NHEJ relative to the WT. The boxes represent mean; whiskers, minimum and maximum values. 'LOD' denotes below the level of detection compared with the WT. The data are presented as the mean ± SEM of four independent experiments. n.s.: not significant, *p < 0.05, **p < 0.005, ****p < 0.0001 versus control. The exact p-values are presented in *Supplementary file 1c*.

The online version of this article includes the following source data and figure supplement(s) for figure 7:

**Figure supplement 1.** Cell cycle analysis of wild-type and *rev7Δ* cells.

**Source data 1.** Raw data corresponding to *Figure 7*, panel B.

the efficiency of plasmid-based NHEJ repair was analyzed by transforming BamHI-linearized plasmid pRS416, which has no homology with the genomic DNA, into the WT and isogenic mutant strains as previously described (*Boulton and Jackson, 1998*; *Moreau et al., 1999*; *Figure 7A*). Consistent with a prior study (*Boulton and Jackson, 1998*), we found that NHEJ was undetectable in the *mre11Δ* mutant (*Figure 7B*, *Supplementary file 1c*). Notably, while the *rev1Δ and rev3Δ* strains showed a modest decrease in the efficiency of NHEJ, *rev7Δ* single and *rev1Δ rev3Δ* double mutants exhibited about four- and twofold decrease, respectively, compared with the WT. Further analysis revealed that the NHEJ efficiency in double mutants – *rev1Δ rev7Δ*, *rev3Δ rev7Δ*, and *sae2Δ rev7Δ* – was comparable to that of *rev7Δ* mutant (*Figure 7B, C —Supplementary file 1c*). Intriguingly, however, we found that ScRev7-42 aa peptide, but not the Rev7-C1 variant, fully restored the NHEJ efficiency in *rev7Δ* cells to the WT levels. It remained possible that the significant reduction in NHEJ efficiency observed in the *rev7Δ* cells could be due to aberrant cell cycle progression. To test this possibility, cell cycle progression in both WT and *rev7Δ* cells was monitored using a fluorescence-activated cell sorter

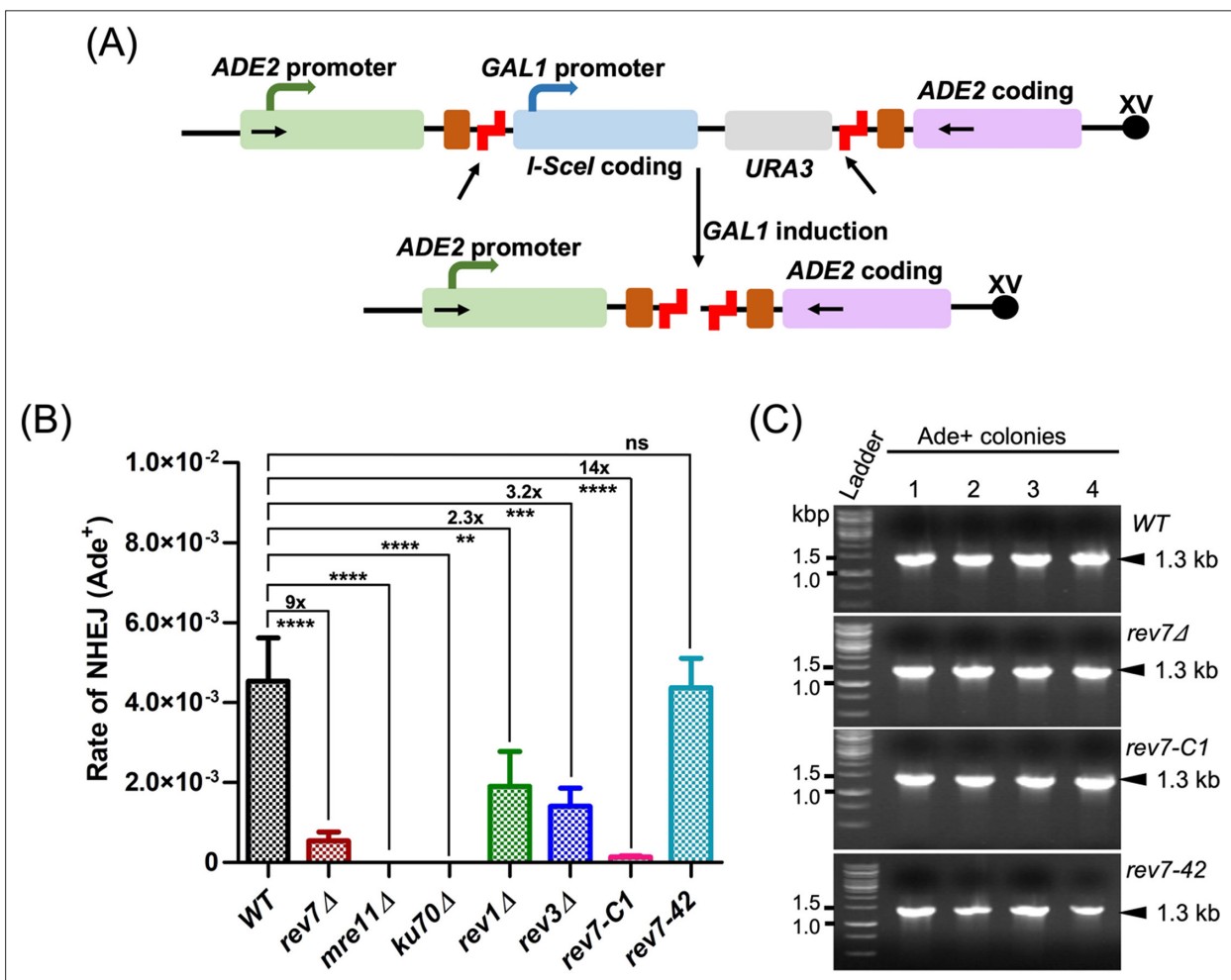

**Figure 8.** *S. cerevisiae REV7* facilitates chromosomal double-strand break (DSB) repair via non-homologous end-joining (NHEJ) pathway. (**A**) A schematic diagram showing the 'suicide deletion' cassette at the *ADE2* locus of chromosome XV redrawn from *Karathanasis and Wilson, 2002*. The arrows indicate the locations of PCR primers. Brown and staggered red boxes correspond to direct repeats and I-SceI cleavage sites, respectively. (**B**) Quantification of NHEJ efficiency in the wild-type YW714 and derivative mutant strains. (**C**) Representative gel images of PCR-amplified DNA products from Ade2+ transformant cells. Data are means ± SEM from three independent experiments. n.s.: not significant, **p < 0.005, ***p < 0.001, ****p < 0.0001 versus control, as assessed by one-way ANOVA Dunnett's multiple comparison test. The exact p-values are presented in *Supplementary file 1d*.

The online version of this article includes the following source data for figure 8:

**Source data 1.** Raw data corresponding to *Figure 8*, panel B.

**Source data 2.** Original file and labelled PDF file for gel images displayed in *Figure 8*, panel C.

(FACS). Compared with the WT cells, we observed a slightly delayed cell cycle progression with *rev7Δ* cells at 30 and 45 min after release from G1 arrest, and there were no differences in their mating type phenotypes. However, after 45 min, *rev7Δ* cells exhibited a similar distribution of cells in the G1, S, and G2 phases of the cell cycle as observed in WT cells (*Figure 7—figure supplement 1*). Together, these results indicate that *rev7Δ* cells do not possess aberrant cell cycle or mating type defects as compared with the WT cells.

For further insights, we sought to determine whether ScRev7 plays a role in NHEJ pathway at the chromosome level. To address this question, the efficiency of NHEJ was evaluated using a 'suicide-deletion' assay in which the I-SceI-induced DSB can be repaired via NHEJ (*Karathanasis and Wilson, 2002*). Briefly, it is based on an approach wherein galactose-induced I-SceI endonuclease inflicts a pair of site-specific DSBs, resulting in the deletion of its own coding region, thereby facilitating the repair of DSB via NHEJ (*Figure 8A*). Using this approach, we determined the frequency of Ade2⁺ recombinants in the WT and isogenic mutant strains. A critical NHEJ factor Ku70 was used as a control. As expected, while NHEJ was undetectable in strains lacking *MRE11* and *KU70*, deletion of *REV7*, *REV1*, and *REV3* led to a 9-, 2.3-, and 3.2-fold decrease, respectively, in the frequency of NHEJ compared with the WT strain. Of note, a 14-fold decrease in the frequency of Ade2⁺ recombinants was observed in the *rev7-C1* cells, which could be restored to WT levels by expressing the C-terminal 42-amino acid peptide (*Figure 8B*, *Supplementary file 1d*). These results reinforce the notion that Rev7 promotes NHEJ repair at DSBs. The PCR product derived from genomic DNA of Ade2⁺ recombinants showed a 1.3-kb amplicon, suggesting faithful repair of I-SceI-induced DSB (*Figure 8C*).

## REV7 plays an anti-recombinogenic role during HR in *S. cerevisiae*

As mentioned above, current evidence suggests that the RIF1/REV7/Shieldin complex blocks DNA end resection and BRCA1-mediated HR, but promotes DSB repair through NHEJ in cancer cells (*Gómez-Llorente et al., 2013*; *Xu et al., 2015*; *Boersma et al., 2015*; *Mirman et al., 2018*; *Findlay et al., 2018*; *Gupta et al., 2018*; *Ghezraoui et al., 2018*; *Dev et al., 2018*; *Tomida et al., 2018*; *Gao et al., 2018*; *Noordermeer et al., 2018*; *Liang et al., 2020*; *Malik et al., 2020*; *Du Truong et al., 2021*). A flurry of research on Shieldin complex in human cancer cells from other laboratories soon followed (*Clairmont and D'Andrea, 2021*). However, the generality of these findings has remained elusive. Therefore, we leveraged a spot assay (*Paeschke et al., 2013*) to understand whether *REV7* plays a role in the regulation of HR in *S. cerevisiae*. In this assay, under conditions of optimal growth and replication stress, we measured the frequency of HR between the *ura3-1* allele on chromosome V and *ura3-G4* allele (*ura3* interrupted by G4 motifs) on the pFAT10-G4 plasmid (*Figure 9A*). In the absence of HU, while no Ura3⁺ papillae were observed in the *ura3-1* strain carrying the empty vector (*Figure 9B*, top row), all other strains harboring the plasmid pFAT10-G4 formed Ura3⁺ papillae (*Figure 9B*).

We sought to build on these observations by testing the effect of HU on the formation of Ura3⁺ papillae by the strains (carrying pFAT10-G4 plasmid) used in the experiment. This analysis revealed that the *ura3-1 rev7Δ* double mutant strain formed significantly more Ura3⁺ papillae than *ura3-1* strains, but less than what was seen in the absence of HU. As anticipated, we observed that both *mre11-D56N,H125N* (used as an internal control) and *rev7Δ mre11-D56N,H125N* mutant strains did not form Ura3⁺ papillae (*Figure 9B*). A similar analysis showed that *ura3-1* strain and the same strain expressing ScRev7-42 amino acid peptide, but not Rev7-C1 variant, displayed a very few, and small Ura3⁺ papillae. As expected, Ura3⁺ papillae formation was not observed in the *ura3-1* strain carrying the empty vector. Quantification indicated that the *ura3-1 rev7Δ* mutant showed 11.5-fold increase in the formation of Ura3⁺ papillae as compared with *ura3-1* strain (*Figure 9C*). Thus, we envision that the absence of Ura3⁺ papillae in the *ura3-1* strain (with or without empty vector) in the presence of HU might be related to Rev7-mediated suppression; whereas their absence in cells that lack functional Mre11 nuclease could be due to HU-induced toxicity. Such an effect has been described previously in the Mre11 nuclease-deficient cells (*Tittel-Elmer et al., 2009*; *Hamilton and Maizels, 2010*). Overall, these results support a model in which *REV7* gene product plays an anti-recombinogenic role during HR, thus its deletion allows cells to facilitate HR between *ura3-G4* and *ura3-1* mutant alleles located on the plasmid and chromosome V, respectively.

To confirm whether the G-quadruplex motifs stimulate HR in the *rev7Δ* strain, the frequency of HR was measured using pFAT10-G4-mut plasmid, which harbors mutations within the G4 forming motifs

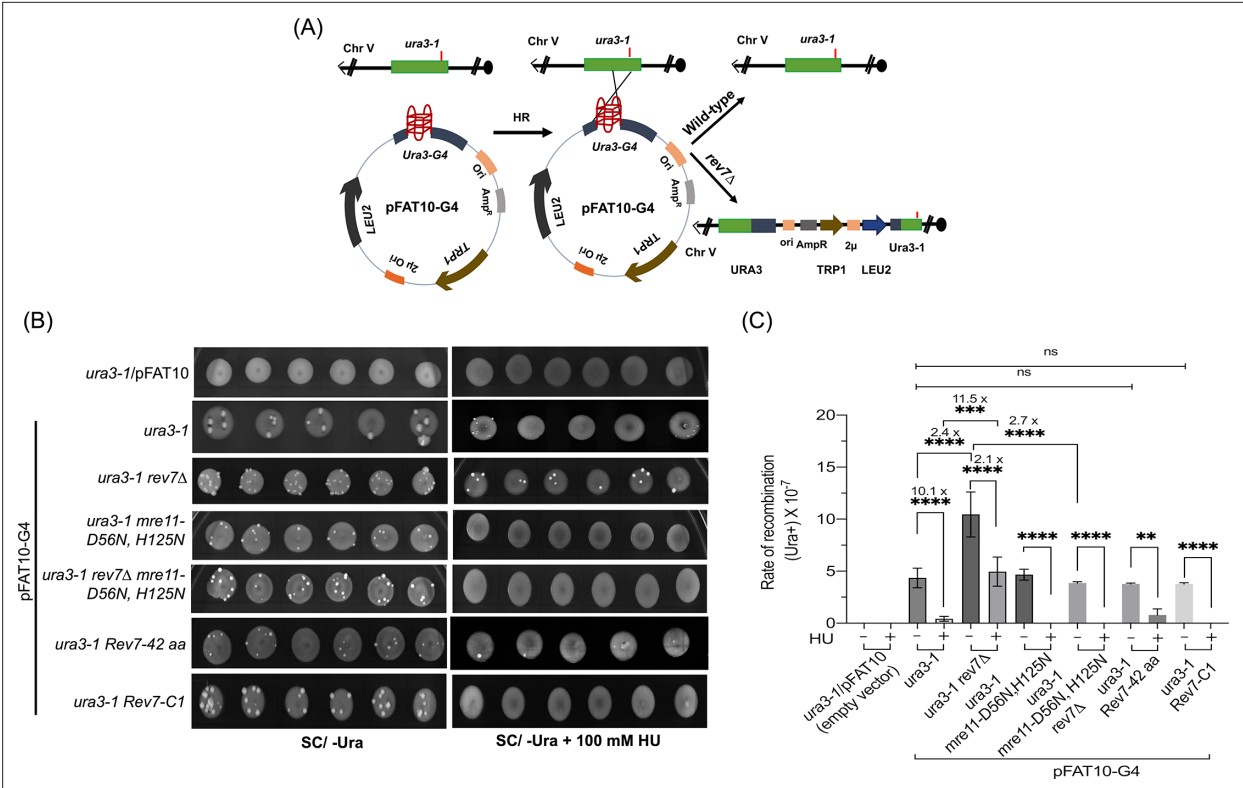

**Figure 9.** Deletion of *REV7* increases the frequency of mitotic homologous recombination (HR). (**A**) Schematic representation of plasmid-chromosome recombination assay. The *ura3-1* and *ura3-G4* alleles are located on chromosome V and plasmid pFAT10-G4, respectively. Recombination between a plasmid borne *ura3-G4* allele and the chromosomal borne *ura3-1* allele would result in Ura⁺ prototrophs. (**B**) Representative images of Ura3⁺ papillae on SC/-Ura agar plates in the absence or presence of 100 mM HU. (**C**) Quantification of the rate of HR frequency in different strains. Data are presented as mean ± SD from three different experiments. ns, not significant, **p < 0.01, ***p < 0.001, ****p < 0.0001 versus control, as assessed using one-way ANOVA and Tukey's post hoc test.

The online version of this article includes the following source data and figure supplement(s) for figure 9:

**Figure supplement 1.** Mutation of G-quadruplex-forming motifs markedly attenuate the rate of homologous recombination (HR) frequency in *rev7Δ* cells.

**Figure supplement 2.** G-quadruplex-forming motifs are stable in *rev7Δ* cells during the homologous recombination (HR) assay.

**Figure supplement 2—source data 1.** Original file and labeled PDF file for the gel image displayed in *Figure 9—figure supplement 2*.

**Figure supplement 3.** Deletion of Rev7 enhances the speed of short-range end-resection in *S. cerevisiae*.

of the *ura3-G4* allele. The results revealed that the frequency of HR reduced by about 5.8-fold, as compared to the strain carrying the plasmid pFAT10 with unmutated G4-forming motifs in the *ura3-G4* allele, revealing that reduction in HR may be caused by mutations in the G-quadruplex-forming motifs (*Figure 9—figure supplement 1*). To assess whether increased frequency of HR is due to the instability of G-quadruplex DNA in *rev7Δ* cells, the length of G4 DNA inserts was assessed in the plasmids isolated from WT and *rev7Δ* cells. The results showed a DNA fragment of 829 bp corresponding to the expected size in both WT and *rev7Δ* cells (*Figure 9—figure supplement 2*), raising the possibility that increased frequency of HR in *rev7Δ* cells could be due to the loss of Rev7 function and instability of G-quadruplex-forming motifs.

Finally, a qPCR-based assay (*Mimitou and Symington, 2010*; *Ferrari et al., 2018*) was leveraged to quantify the amounts of ssDNA generated in the *rev7Δ* and *rad51Δ mre11-H125N rev7Δ* strains at a HO endonuclease-induced DSB (*Figure 9—figure supplement 3A*). The results indicated a significant increase in the percentage of ssDNA 0.7 kb distal to the DSB at 4 hr after its induction in the *rev7Δ rad51Δ* cells compared with *rad51Δ* cells (*Figure 9—figure supplement 3B*). As expected, the DNA end resection rates in the *mre11-H125N* and *mre11-H125N rev7Δ* cells were similar to that in the *rad51Δ* cells. Importantly, at 3 kb distal to the DSB, *rev7Δ* cells showed comparable rates of

DNA end resection, in line with both *rad51Δ* and *mre11-H125N* cells (**Figure 9—figure supplement 3C**). Overall, these results support the notion that Rev7 suppresses HR by inhibiting Mre11-mediated short-range DNA end resection.

## Discussion

In this study, we reveal the surprising finding that Rev7 physically associates with the subunits of the MRX complex and also provide unanticipated insights into the mechanism by which it regulates the DSB repair pathway choice between HR and NHEJ in *S. cerevisiae*. Notably, we demonstrate that Rev7 binds to the subunits of the MRX complex, via a 42-residue C-terminal segment (residues 203–245), protects cells form G4 DNA-HU-induced toxicity, facilitates DSB repair via NHEJ while blocking HR. In addition, we present robust evidence that ScRev7 impedes Mre11 nuclease and Rad50's ATPase activities, without affecting the latter's ability to bind ATP. When seen from a teleological perspective, our work is conceptually reminiscent of Shieldin complex-mediated suppression of 5′ end resection and repair of DSBs via NHEJ by blocking HR in human cancer cells. It remains plausible that this alternative mechanism of DSB repair pathway choice in *S. cerevisiae* might be conserved across multiple species.

Historically, *REV7* was discovered as playing an important role in DNA damage-induced mutagenesis in *S. cerevisiae* (**Lemontt, 1971**; **Lawrence et al., 1985a**). Further investigations showed that Rev7 associates with Rev3 and functions as a regulatory subunit of eukaryotic TLS DNA polymerase Pol ζ (**Prakash et al., 2005**; **Maiorano et al., 2021**; **Ling et al., 2022**; **Paniagua and Jacobs, 2023**). While Rev7 has no known enzymatic activity, it acts as a versatile scaffolding protein with pleiotropic functions in diverse cellular processes including, but not limited to, epigenetics, immune signaling, viral mutagenesis and cancer development that were initially considered inconceivable (**Decottignies, 2013**; **de Krijger et al., 2021**). However, little is known about the nature of specific effector(s) that associate with Rev7 and regulate DSB repair pathway choice in *S. cerevisiae*. As we discuss below, our work convincingly demonstrates that ScRev7 physically interacts with the individual subunits of MRX complex and regulates their activities. In line with this, MST-based protein–protein interaction assays, supported by AF2-multimer modeling, imply tight association between the Mre11 and Rad50 subunits and ScRev7. It is interesting to note that Sae2, which cooperates with the MRX complex to initiate DNA end resection, does not associate with Rev7, suggesting interaction specificity between MRX subunits and Rev7.

However, an intriguing inquiry arises regarding whether the MRX subunits exist as separate entities in the cell. Although, to our knowledge, the existence and relative amounts of monomeric, dimeric and trimeric species of MRX subunits in vivo are unknown, current evidence suggests that both Mre11 and Rad50 subunits independently bind DNA (**Stracker and Petrini, 2011**; **Paull, 2018**; **Casari et al., 2019**). Other studies have shown that MRX complex binds and tethers DNA ends, which is thought to be required for DSB repair (**Trujillo et al., 2003**; **Cassani et al., 2016**). Furthermore, while data obtained from co-immunoprecipitation experiments implicate that Mre11, Rad50, and Xrs2 subunits exist as a hetero-trimeric complex in vivo (**Usui et al., 1998**), in vitro experiments have shown the formation of dimeric Mre11–Rad50 and trimeric MRX complexes (**Oh et al., 2016**; **Arora et al., 2017**). Collectively, these results allow us to postulate that the monomeric, dimeric, and trimeric species of MRX subunits might exist in a dynamic equilibrium under in vivo conditions.

Given that Rev7 and MRX complexes play essential roles in DNA repair pathways, our findings provide novel insights into how ScRev7 interacts with the MRX subunits and regulates their functions. Of note, structure–function analyses showed that deletion of 42-amino acid segment (203–245) at the extreme C-terminus of ScRev7 abolished its ability to interact with the MRX subunits, whereas loss of the N-terminal HORMA domain, an evolutionarily conserved protein–protein interaction module (**Muniyappa et al., 2014**; **Rosenberg and Corbett, 2015**; **de Krijger et al., 2021**) had no discernible effect on their interactions. Curiously, further analysis indicated that the 42-amino acid peptide alone was sufficient to bind to the subunits of MRX complex and regulate the DSB repair pathway choice between NHEJ and HR. Reciprocally, future work will be required to determine the regions/domains of MRX subunits that interact with ScRev7 for a comprehensive understanding of the crosstalk between these components. Work is currently in progress to gather insights into these questions.

The data from MST-based protein–protein interaction assays informed that ScRev7 binds to the MRX subunits with sub-micromolar affinity, analogous to the interaction between HORMA-domain protein hMAD2 and hCDC20 (**Piano et al., 2021**). How might ScRev7 attenuate the function of Mre11

and Rad50 subunits? Our finding indicate that nanomolar amounts of ScRev7 besides attenuating the Mre11 nuclease activity also impedes Rad50's ATPase activity without obstructing the ability of the latter to bind ATP. While the specific nuances of the interaction await further research, we surmise that physical interaction between ScRev7 and the Mre11/Rad50 subunits might contribute to the observed effects. Although we provide compelling Y2H data that the C-terminal 42-amino acid peptide of Rev7 is critical for binding to the MRX subunits and protect cells from G4 DNA-HU-induced toxicity, and regulate the pathway choice between NHEJ and HR; however, we could not demonstrate its function in vitro because of technical difficulties associated with its expression and purification. By comparison, we found that the Rev7-C1 variant, which lacks the ability to interact with the MRX subunits in Y2H assays, was capable of blocking, albeit partially, the catalytic activities of Mre11 and Rad50 subunits in vitro. A parsimonious explanation would be that amino acid residues in the Rev7-C1 fragment might be involved in pairwise interactions between ScRev7 and the Mre11 and Rad50 subunits, as seen in AlphaFold2 models. Conversely, the 42-amino acid fragment as a part of whole protein might act as a lid to block the residues in the Rev7-C1 fragment, thereby enabling it to function effectively as a single site for binding to the MRX subunits in Y2H assays. Future studies are required to test this hypothesis.

Many lines of evidence indicate that G-quadruplex-forming motifs are measurably enriched at certain functional regions in the genomes of all organisms, from humans to plants to microbes (*Huppert and Balasubramanian, 2005*; *Capra et al., 2010*; *Castillo Bosch et al., 2014*; *Lejault et al., 2021*). Furthermore, G-quadruplex structures modulate diverse cellular processes, including DNA replication, transcription, translation, and are associated with certain diseases, characterized by high rates of chromosomal instability (*Rhodes and Lipps, 2015*; *Spiegel et al., 2020*). Interestingly, we found that *rev7Δ* mutant cells harbouring exogenous G-quadruplex-forming motifs exhibit hypersensitivity to HU-induced genotoxic stress and cell death. This is consistent with emerging evidence that DNA replication stress induced by G-quadruplex structures play a prominent role in triggering genomic instability, which is exacerbated in the presence of HU (*Sato and Knipscheer, 2023*). Although the precise mechanism remains unclear, it is tempting to speculate that Rev7 may recruit G4-resolving helicase(s) such as Sgs1 and Pif1 to unwind G-quadruplex structures prior to or during DNA replication (*Huber et al., 2002*; *Paeschke et al., 2013*). Alternatively, or in addition, it might generate G4 DNA intermediates which may be processed or cleaved by specific enzymes, including Mus81, Sgs1, or Mre11 (*Regairaz et al., 2011*; *Sun et al., 1999*; *Ghosal and Muniyappa, 2005*; *Ghosal and Muniyappa, 2007*). Regardless, our data align with the notion that G-quadruplex structures are endogenous sources of replication stress and their formation and persistence may lead to genomic instability and cell death.

As mentioned above, Shieldin complex facilitates NHEJ-dependent DSB repair, while inhibiting DNA end resection and HR in cancer cells (*Clairmont and D'Andrea, 2021*; *Paniagua and Jacobs, 2023*). Since 5′ end resection is the primary step in HR-mediated DSB repair, we hypothesized that physical interaction between ScRev7 and MRX subunits might block resection of DNA termini, and then facilitate NHEJ-dependent DSB repair instead of HR. Consistent with this premise, we found that ScRev7 promotes NHEJ by blocking HR. However, other regulatory components may also play a role in modulating the levels of NHEJ versus HR-mediated DSB repair. For instance, TRIP13 or p31[comet] inhibits the interaction of Rev7 with the SHLD3 subunit and regulates DNA end resection at DSBs and promotes their repair by HR (*Sarangi et al., 2020*). Additionally, interaction between Rev7 and putative binding partners may be regulated by posttranslational modifications and chromatin accessibility under changing physiological conditions.

Although the mechanism underlying Rev7-mediated regulation of DSB repair between *S. cerevisiae* and human cancer cells indicate broad similarities, specific variations exist, including striking differences. A fundamental difference between the human cancer cells and *S. cerevisiae* is that, while ScRev7 robustly interacts with and suppresses the biochemical activities of both Mre11 and Rad50 subunits to facilitate NHEJ, Rev7–Shieldin complex acts as a downstream effector of 53BP1-RIF1 in restraining DNA end resection to promote NHEJ (*Clairmont and D'Andrea, 2021*; *Paniagua and Jacobs, 2023*). Furthermore, hRev7 interacts with the SHLD3 subunit via the HORMA domain, which is entirely dispensable in the case of ScRev7. Thus, we posit that the Shieldin complex-mediated regulation of DSB repair pathway choice might be a source of evolutionary innovation as an additional layer of regulation. In summary, our data provide novel insights into the alternative mechanism underlying the regulation of DSB repair pathway choice in *S. cerevisiae*.

# Materials and methods

## S. cerevisiae strains, DNA plasmids and oligonucleotides

All strains and primers used for the construction of strains/plasmids used in this study are listed in **Supplementary file 1e and f**, respectively. The plasmids FT10/Chr.IV$_{G4}$_lg and FT10/Chr.IV$_{G4}$_le were a kind gift from Dr. Virginia Zakian.

## Construction of strains used in the study

The *S. cerevisiae* W1588a haploid strains carrying single/double deletions in the genes *rev1Δ*, *rev3Δ*, *rev7Δ*, *rev1Δ rev3Δ*, *rev7Δ mre11Δ*, or *sae2Δ rev7Δ* were constructed using appropriate pairs of primers as previously described (**Sambrook and Russell, 2001**; **Janke et al., 2004**). The *REV7* gene was deleted using the KanMX4 (pFA6a- KanMX4) cassette utilizing the forward primer OSB11 and reverse primer OSB12, and deletion was confirmed by PCR using gene-specific primers OSB13 and OSB14. Similarly, other strains were generated as follows: *MRE11* was deleted using the hphNT1 cassette (pYM-hphNT1) utilizing the OSB52 and OSB54 primers; deletion was verified by PCR using primer OSB53. The *REV1* was deleted in a similar fashion, using the hphNT1 cassette and the OSB55 and OSB56 primers; deletion was ascertained by PCR using primer OSB57. *REV3* gene was deleted using the KanMX4 cassette (derived from pFA6a-hphNT1, pFA6a-KanMX4, respectively) and the primers OSB58 and OSB59; deletion was confirmed by PCR using primer OSB60. The hphNT1 (pYM-hphNT1) cassette was used in the generation of double mutants (listed in **Supplementary file 1f**). The *rev7-C1* and *rev7-42* mutant strains were constructed by inserting DNA sequences encoding N- and C-terminally truncated forms of Rev7 at the endogenous loci using *rev7-C1*-9MYC-hphNT1 and *rev7-42*-3MYC-KANMX4 cassettes via overlapping primer-based PCR. The *rev7-C1*-9MYC-hphNT1 cassette was generated using primer pairs OSB116, OSB117 and OSB118, OSB70. Likewise, *rev7-42*-3MYC-KANMX4 cassette was generated using primer pairs OSB122, OSB120 and OSB121, OSB70. The N- and C-terminal truncation variants were confirmed by PCR using probes OSB13 and OSB14, respectively.

The *mre11Δ*, *rev1Δ*, *rev3Δ*, and *rev7Δ* mutant strains were generated in the strain YW714 using the hphNT1 cassette (derived from pFA6a-hphNT1). The *REV7* truncation variants with the c-myc tag sequence – *rev7-C1-9MYC* and *rev7-42* aa-3MYC – were generated as described above. The *REV3* gene was deleted using the hphNT1 cassette in the strain PJ694A utilizing primers OSB58 and OSB59 and deletion was confirmed using primer OSB60. Likewise, *mre11Δ rad50Δ xrs2Δ* triple mutant was generated in the strain PJ694A as follows: *mre11Δ::KANMX4* was generated by PCR amplification of KANMX4 cassette (pFA6a-KANMX4) using forward and reverse primers, OSB52 and OSB53, respectively. Gene deletion was confirmed by PCR using primer OSB54. The *rad50Δ::URA3* strain was constructed by PCR amplification of URA3 cassette (pAG60-URA3) using primers OSB133 and OSB75; deletion was confirmed using primer OSB76. Similarly, *xrs2Δ::HphNT1* was generated by PCR amplification of hphNT1 cassette (pFA6a-hphNT1) using primers OSB134 and OSB78, and deletion was confirmed using primer OSB79. The *rev7Δ* mutants were generated in the strain LSY2172-24C and LSY2265-10D, as described above.

## Construction of DNA plasmids for expression and recombination assays

The plasmids FT10/Chr.IV$_{G4}$_lg and FT10/Chr.IV$_{G4}$_le were a kind gift from Dr. Virginia Zakian. *S. cerevisiae REV7* gene was amplified from genomic DNA by PCR using primer pair (forward OSB01 and reverse OSB02) and Phusion DNA polymerase (NEB, Ipswich, MA) as previously described (**Sambrook and Russell, 2001**). The reaction yielded an amplicon of expected size, which was digested with BamHI/HindIII, and ligated into a BamHI/HindIII digested pET28a(+) vector (Novagen) using T4 DNA ligase. The resulting expression plasmid was designated pET-28a_REV7. Analogously, the truncated plasmid pET28a_Sc*REV7*-C1 was generated by PCR amplification using primers OSB01 and OSB125 and pET28a_Sc*REV7* plasmid DNA as a template. The PCR product was digested with BamHI/HindIII and ligated into BamHI/HindIII digested pET28a vector. Likewise, *S. cerevisiae RAD50* gene was PCR-amplified from genomic DNA using OSB33 (forward) and OSB34 (reverse) primers. The amplicon was digested with BamHI/Xho1 and ligated into a BamHI/Xho1 digested pE-SUMO Kan vector (Life Sensors, Malvern, PA) using T4 DNA ligase. The resulting expression plasmid was designated pE-SU-MO_RAD50. The REV7-eGFP expression vector was constructed by PCR amplification of *S. cerevisiae REV7* gene in the pET-28a_REV7 plasmid using forward Rev7-eGFP primer and reverse Rev7-eGFP

primer (*Supplementary file 1f*). The amplicon of expected size was digested with XbaI/XhoI and ligated into XbaI/XhoI digested pPROEX vector, upstream of eGFP coding sequence. The resulting plasmid encodes ScRev7-eGFP fusion protein. Similarly, the DNA sequence encoding ScRev7-C1 was amplified from pET28a-REV7 plasmid using primers as shown in *Supplementary file 1f*. The amplicon was then cloned into pPROEX::eGFP vector. The pPROEX::eGFP vector was a kind gift from Deepak Saini.

The *ura3-G4* insert was amplified by overlap extension PCR method using OSB80 and OSB82 as forward and OSB81 and OSB83 as reverse primers, respectively. The amplicon was digested with BamHI/SphI and ligated into a BamHI/SphI digested FAT10 vector. The primer OSB82 corresponds to the DNA sequence 362751–362775 of the coding strand of *S. cerevisiae* Chr X. The primer OSB81 is its complementary strand. Similarly, *ura3-G4* mutant was generated using OSB80 and OSB130 as forward, and OSB129 and OSB83 as reverse primers. The PCR product corresponding to full-length *ura3-G4* mutant was digested with BamHI/Sph1 and cloned into FAT10 vector. The oligonucleotide (ODN) OSB129 sequence 362751–362775 corresponds to mutant version of *S. cerevisiae* Chr X. The OSB130 is its complementary strand.

## Construction of DNA plasmids for yeast two-hybrid analysis

To construct the pGBKT7-*REV7,* the *S. cerevisiae REV7* orf in the pET28a (+) *REV7* construct was PCR-amplified using Phusion DNA polymerase and forward OSB03 and reverse OSB04 primers, respectively. The amplicon was digested with Nde1/BamHI and ligated into Nde1/BamHI digested pGBKT7 vector. The *S. cerevisiae MRE11* gene was PCR-amplified from the genomic DNA using OSB05 forward primer and *MRE11*_RP reverse primer. The amplicon was digested with BamHI/EcoRI and ligated into the pGADT7 prey expression vector at the BamHI/EcoRI site. The same procedure was leveraged for the construction of all other prey expression vectors: *XRS2* was amplified from the genomic DNA using forward OSB36 and reverse OSB37 primers. The amplicon was digested with Nde1/BamHI and ligated into the pGADT7 prey expression vector at the NdeI/BamHI site; *RAD50* was amplified from the pESUMO_*RAD50* construct using forward OSB35 and reverse OSB34 primers. The amplicon was digested with EcoRI/Xho1 and ligated into the pGADT7 prey expression vector digested with EcoRI and XhoI; *REV7* was amplified from the pET-28a_*REV7* construct using the same primers and ligated into pGADT7 prey expression vector as in the case of bait plasmid construction. The *SAE2* gene was amplified from the plasmid pET21a-*SAE2* (*Ghodke and Muniyappa, 2013*) using primers OSD150 and OSD151 and the amplicon was digested with NdeI and EcoRI and cloned into pGADT7 vector.

The N-terminally truncated ScREV7 variants (REV7-N1, REV7-N2, and REV7-N3) were constructed by PCR amplification of relevant portions of the *REV7* gene using OSB61, OSB62, OSB63 as a forward primers and OSB4 as a reverse primer (common for all N-terminal deletions), respectively. The C-terminally truncated ScREV7 variants (REV7-C1, REV7-C2, and REV7-C3) were constructed via PCR amplification of relevant portions of the *REV7* gene in the pET-28a_*REV7* plasmid using OSB3 as a forward primer (common for all C-terminal deletions), and OSB64, OSB65, and OSB66 as reverse primers, respectively. The amplicons, corresponding to the N- and C-terminal variants, were digested with NdeI/BamHI and ligated into NdeI/BamHI digested pGBKT7 vector. *S. cerevisiae* PJ694A strain was used in Y2H analyses.

## Cell viability assay

The *S. cerevisiae* WT and isogenic mutant strains were grown in liquid YPD or SC medium to an $OD_{600}$ of 0.5. Tenfold serial dilutions were spotted on YPD agar plates with or without HU. Similarly, cells were spotted on SC selection medium plates lacking the indicated amino acids, with or without inhibitors, as indicated in the figure legends. The plates were incubated at 30°C for 3–4 days. The images were captured using epi-illumination at auto-exposure ChemiDoc MP imaging system (Bio-Rad).

## Yeast two-hybrid interaction analysis

The Y2H assays were performed as previously described (*Fields and Song, 1989*; *Thakur et al., 2020*) Briefly, pairwise combination of plasmids expressing bait proteins, fused to the Gal4 DNA-binding domain (G4BD), and prey proteins, fused to the Gal4 activation domain (G4AD), were co-transformed into the WT strain PJ69-4A, *rev3*Δsingle and *mre11Δ rad50Δ xrs2Δ* triple mutant strain. The empty prey vector and a vector expressing Rev7 served as negative and positive controls, respectively. The

interactions were analyzed by spotting the indicated transformants on SC/-Trp-Leu and SC/-Trp-Leu-His agar plates containing 15 mM 3-AT, which were then incubated for 3–5 days at 30°C. Growth of cells on SC-Leu-Trp-His+3-AT agar plates is indicative of moderate/strong protein–protein interactions between the bait and prey proteins.

## Assay for G-quadruplex DNA-HU-induced toxicity

The assay was performed as previously described (*Paeschke et al., 2011*). The plasmid constructs used in this study were identical to the parent plasmid, pFAT10, except that its derivatives contained three tandem repeats of guanine residues derived from chromosome IV, inserted in the lagging or leading templates. Briefly, the *S. cerevisiae* WT, *rev7Δ* mutant, and cells expressing ScRev7 truncation variants (ScRev7-C1, ScRev7-42 aa) harboring the pFAT10 plasmid were synchronized in the G1 phase using α-factor at 30°C and then released from the pheromone block by washing the cells. Subsequently, the cultures were diluted to yield identical $A_{600}$ values. Tenfold serial dilutions of each culture was spotted on SC/-Leu or SC/-Leu/+100 mM HU agar plates, which were incubated at 30°C for 72 hr.

## Expression and purification of ScRev7

The *S. cerevisiae* Mre11, Rad50, Xrs2, and Sae2 proteins were expressed and purified as previously described (*Ghosal and Muniyappa, 2007*; *Ghodke and Muniyappa, 2016*). The *S. cerevisiae REV7* gene was sub-cloned into pET28a(+) expression vector with an N-terminal His$_6$-tag. The resulting plasmid was designated as pET28a(+)_REV7. The *E. coli* BL21(DE3)pLysS cells harboring pET28a(+)_REV7 plasmid were grown at 37°C in Luria-Bertani broth (1% tryptone, 0.5% yeast extract, 1% NaCl, pH 7.0) containing 50 µg/ml kanamycin in an orbital shaking incubator at 180 rpm to an OD$_{600}$ of 0.6, and then ScRev7 expression was induced by adding 1-thio-β-D-galactopyranoside (IPTG) to a final concentration of 0.1 mM. Following the addition of IPTG, cultures were grown in an orbital shaking incubator at 25°C for 12 hr. Cells were harvested by centrifugation at 6000 × *g* and the cell paste was resuspended in 50 ml of buffer A (20 mM Tris–HCl (pH 8.0), 150 mM NaCl, 10% glycerol, and 5 mM 2-mercaptoethanol containing 1 mM phenylmethylsulfonyl fluoride and 0.05% Triton X-100). The cells were lysed by sonication on ice (7 × 1 min pulses) and subjected to centrifugation at 30,000 rpm at 4°C for 20 min. Solid ammonium sulfate was added (0.472 gm/ml) to the supernatant with continuous stirring for 45 min at 24°C. The precipitate was collected by centrifugation at 18,000 rpm at 4°C for 1 hr. The precipitate was dissolved in buffer A and dialyzed against the same buffer. The dialysate was loaded onto a 5-ml Ni$^{2+}$-NTA column (QIAGEN, Valencia, CA). After washing the column with buffer A containing 20 mM imidazole, bound proteins were eluted with a gradient of 20→500 mM imidazole. The fractions containing ScRev7 were pooled and dialyzed against buffer B (20 mM Tris–HCl, pH 8.0, 1.2 M NaCl, 7% glycerol, 5 mM 2-mercaptoethanol). ScRev7 protein was further purified by chromatography using a Superdex S75 gel filtration column, attached to an AKTA Prime FPLC system, which had been equilibrated with and eluted using buffer B. The peak fractions containing Rev7 were pooled and dialyzed against buffer C (20 mM Tris–HCl (pH 8.0), 30 mM NaCl, 30% glycerol, 5 mM 2-mercaptoethanol), and loaded onto a heparin column (5 ml). The bound proteins were eluted with a gradient of 30→350 mM NaCl in buffer C. Aliquots of each fraction were analyzed by SDS–PAGE and the protein bands were visualized by staining the gel with Coomassie brilliant blue. The fractions that contained Rev7 were pooled and dialyzed against buffer D (20 mM Tris–HCl (pH 8.0), 100 mM NaCl, 10% glycerol, 1 mM DTT), and stored at −80°C.

## Expression and purification of ScRev7-eGFP and ScRev7C1-eGFP

The His$_6$-tagged ScRev7-eGFP and ScRev7C1-eGFP fusion proteins were expressed in and purified from whole-cell lysates of *E. coli* BL21(DE3)pLysS host strain harboring the pPROEX/REV7-eGFP plasmid as described above. The His$_6$-tagged eGFP was purified from the cell lysates of *E. coli* BL21*(DE3) pLysS host strain harboring eGFP:pPROEX expression plasmid by Ni$^{2+}$-NTA affinity chromatography. Briefly, the whole-cell lysate was loaded onto a 5-ml Ni$^{2+}$-NTA resin column, which had been equilibrated with a buffer A (4-(2-Hydroxyethyl)-1-Piperazine Ethanesulfonic acid [HEPES], pH 7.5, 50 mM NaCl and 10% glycerol). The column was washed with buffer A containing 70 mM imidazole. The bound proteins were eluted with a gradient of 70→800 mM imidazole in buffer A. Aliquots of each fraction were analyzed by SDS–PAGE and the protein bands were visualized by staining the gel with Coomassie brilliant blue. The protein concentrations were determined by the Bradford assay. The

fractions that contained ScRev7-eGFP and eGFP were pooled, dialyzed against buffer A, and stored at −80°C.

## Expression and purification of *S. cerevisiae* Rev1

The GST-tagged Rev1 was purified as described previously with some modifications (*Johnson et al., 2006*). Briefly, the *S. cerevisiae* BJ5464 cells carrying plasmid pBJ842-Rev1-GST were selected on SC plates lacking leucine. Single colonies were inoculated into liquid SC/-Leu medium (200 ml) containing 2% raffinose, and the cultures were grown for 12–14 hr at 30°C until the $A_{600}$ nm reached 0.5. Following this, the cultures were centrifuged at 14,000 rpm for 10 min and the pellet was washed thrice with MilliQ water. Cells were then resuspended in liquid SC/-Leu medium containing 2% galactose. The cultures were grown at 30°C in an orbital shaking incubator at 250 rpm for 7 hr, harvested by centrifugation at 4000 rpm for 10 min, resuspended in CBB buffer (50 mM Tris–HCl pH 7.5, 10% sucrose, 1 mM Ethylenediaminetetraacetic acid (EDTA), 10 mM β-mercaptoethanol and 5 µg/ml of protease inhibitor cocktail). Cells were lysed using a FastPrep 24 homogenizer and the cell debris was separated by centrifugation at 10,000 rpm for 10 min. The supernatant was centrifuged at 35,000 rpm for 30 min. The protein(s) in the supernatant was precipitated using ammonium sulfate (0.208 g/ml), followed by centrifugation at 20,000 rpm for 45 min. The pellet was resuspended in the GST-binding buffer (GBB, 50 mM Tris–HCl pH 7.5, 10% glycerol, 1 mM EDTA, 10 mM β-mercaptoethanol, and 5 µg/ml of protease inhibitor cocktail) and dialyzed extensively against the same buffer. The sample was loaded onto a GST-column (GSTrap High performance column, 5 ml, Cytiva Life Sciences) at a rate of 0.5 ml/min. The column was washed with GBB buffer containing 500 mM NaCl. Bound protein was eluted using GBB buffer containing 40 mM glutathione. Fractions (1 ml each) were collected at a flow rate of 1 ml/min, analyzed on 10% SDS–PAGE and staining the gel with Coomassie brilliant blue. The fractions containing pure Rev1-GST protein were pooled and dialyzed against MST buffer (HEPES, pH 7.5, 50 mM NaCl, 10% glycerol) and stored at −80°C. The concentration of ScRev1 was determined by the dye-binding assay.

## SDS–PAGE and immunoblot analysis

Western blot analysis was performed as previously described (*Mahmood and Yang, 2012*). The *S. cerevisiae* PJ69-4A strains were co-transformed with empty vectors pGBKT7 and pGADT7, or a bait vector harboring Rev7 truncations in combination with prey vectors harboring *REV7*, *MRE11*, *XRS2*, or *RAD50*. The transformants were selected by plating on SC/-Leu-Trp agar medium, and single colonies were inoculated into liquid SC/-Trp-Leu medium (5 ml) containing 2% dextrose. The cultures were incubated at 30°C for 12–14 hr in a rotatory incubator at a speed of 200 rpm to $A_{600 \text{ nm}}$ = 0.15, which were then transferred onto 10 ml of liquid SC/-Trp-Leu medium. Incubation was continued at 30°C with shaking to until the culture reached $A_{600 \text{ nm}}$ = 0.5. The cells were harvested by centrifugation at 4000 rpm for 5 min and the pellet was resuspended in lysis buffer (50 mM sodium-HEPES pH 7.5, 200 mM sodium acetate pH 7.5, 1 mM Ethylenediaminetetraacetic acid (EDTA), 1 mM Ethylene glycol tetraacetic acid (EGTA), 5 mM magnesium acetate and 5% glycerol). Cells were lysed with 2 min bursts at 4.0 m/s on a FastPrep-24 homogenizer in the presence of 200 ml of acid-washed glass beads. The samples were centrifuged at 7000 rpm for 5 min to remove the beads and cell debris. The supernatant was centrifuged at 13,000 rpm for 10 min. Equal amounts of protein (50 µg protein from the supernatants) were separated by 10% SDS–PAGE. The proteins from the gel were transferred onto a polyvinyl difluoride membrane (0.45 µm). The blot was blocked with 5% nonfat dry milk in Tris-buffered saline containing Tween 20 (20 mM Tris–HCl buffer, pH 7.5,150 mM NaCl, 0.1% Tween 20) buffer for 1 hr at 25°C, washed, and probed at 25°C for 1 hr with anti-Myc antibodies (dilution 1:3000 dilution). Subsequently, the blots were washed thrice with TBST buffer and incubated with HRP-conjugated anti-rabbit antibodies (dilution 1:20,000; Sigma-Aldrich) at 25°C for 1 hr. Finally, the blots were washed thrice with TBST buffer and developed using chemiluminescence substrates (Bio-Rad Laboratories, CA, USA) in the Bio-Rad ChemiDoc Imaging systems. Anti-Pgk1 antibodies were obtained from Santa-Cruz Biotechnologies, CA, USA.

## AlphaFold-Multimer predictions of Mre11–Rev7 and Rad50–Rev7 protein complexes

The sequences of full-length ScRev7, Mre11 and Rad50 were obtained from the *Saccharomyces* Genome Database (https://www.yeastgenome.org/). The ScRev7-Mre11 and ScRev7-Rad50 heterodimer models were built using AF2-multimer (*Evans et al., 2022*; *Varadi et al., 2022*) via the ColabFold software Version 1.5.5 (https://github.com/sokrypton/ColabFold; *Mirdita et al., 2022*, RRID:SCR_025453). The models with the highest confidence were analyzed using LigPLot software (Version 2.2) to identify the amino acid residues involved in the interactions between the Mre11/Rad50 subunits and ScRev7. Approximately, 44% and 43% of the residues across the binding interface of ScRev7 and Mre11 displayed confident pLDDT scores greater than 60. Similarly, 67% and 100% of the residues lining the ScRev7 and Rad50 interface exhibited pLDDT scores greater than 60. The PyMOL Molecular Graphics System (Version 2.5.5) was used to visualize and analyze protein structures. Among the five different types of models generated by AlphaFold-Multimer, the top models (based on average pLDDT score) were chosen for display.

## MST assay

The MST assays were carried out on a Monolith NT.115 instrument (NanoTemper Technologies GmbH) according to the manufacturer's instructions. Samples were prepared in 20 µl MST buffer (HEPES, pH 7.5, 50 mM NaCl, 10% glycerol) containing ScRev7-eGFP or eGFP and different concentrations of ligands in the range: Mre11 (0.00015–5 µM), Rad50 (0.00006–2 µM), Xrs2 (0.00015–5 µM), Mre11-Rad50 (0.00015–5 µM), Rev1 (0.00015–5 µM), or Sae2 (0.00015–5 µM). After incubation at 37°C for 15 min, samples were transferred into Monolith NT.115 glass capillaries. The measurements were performed using 40% MST power with laser on/off times of 30 and 5 s, and the MST signals were normalized to fraction bound (X) by $X = [Y(c) - Min]/(Max - Min)$, error bars (SD) were normalized by $stdnorm = std(c)/(Max - Min)$. The Fnorm values or fraction bound were plotted against the ligand concentration to obtain an estimate of binding affinity. The values of the Hill coefficient and equilibrium dissociation constants ($K_d$) were calculated using the isothermal binding equation model in the MO. Affinity Analysis software provided by NanoTemper. Statistical analysis of data was performed using GraphPad Prism software (v5.0).

## Preparation of radiolabeled DNA substrates

The sequences of ODNs used for the preparation of DNA substrates are listed in *Supplementary file 1g*. The ODNs were labeled at the 5′ end using $[\gamma\text{-}^{32}P]ATP$ and T4 polynucleotide kinase, as previously described (*Sambrook and Russell, 2001*). The unincorporated $[\gamma\text{-}^{32}P]ATP$ was removed using a Sephadex G-50 superfine mini-column. The dsDNA substrates were prepared as follows: 41 bp dsDNA by mixing aliquots of 5′-end $^{32}P$-labeld OSB17 (upper strand) with a small excess of unlabeled complementary OSB20; 60 bp dsDNA by mixing OSB41 (upper strand) and OSB42 (lower strand) in 1× Saline-Sodium Citrate buffer (0.3 M sodium citrate buffer, pH 7.0, containing 3 M NaCl), followed by heating at 95°C for 5 min and then slowly cooling to 24°C over a period of 90 min. The substrates were resolved by non-denaturing 8% PAGE for 4 hr at 4°C (*Thakur et al., 2020*). The substrates were eluted from the gel slices in TE buffer (10 mM Tris–HCl, pH 7.5, and 1 mM EDTA) and stored at 4°C for further use.

## Exonuclease assay

The assay was performed as previously described (*Arora et al., 2017*) with slight modifications. Two steps are involved in the assay: the first step involves the assembly of the ScRev7–MRX complex, and the second, DNA cleavage. The MRX subunits (100 nM each) were mixed with increasing concentrations of ScRev7 (0.05–2.5 µM) or ScRev7-C1 (1.5 and 3 µM) and incubated on ice for 15 min, prior to the addition of twenty µl of reaction mixture that contained 25 mM MOPS (pH 7.0), 20 mM Tris–HCl (pH 7.5), 80 mM NaCl, 8% glycerol, 5 mM $MnCl_2$, 5 mM $MgCl_2$, 1 mM ATP, 1 mM DTT, 200 µg/ml BSA, and 10 nM of $^{32}P$-labeled 60 bp dsDNA. Subsequently, reaction mixtures were incubated at 37°C for 1 hr, and the reaction stopped by adding a two µl solution containing two mg/ml proteinase K, 50 mM EDTA and 1% SDS. After incubation at 37°C for 30 min, 6 µl of a solution containing 10 µg/ml glycogen, two µl 3 M sodium acetate, and 50 µl of absolute ethanol was added to each sample and were frozen at −80°C. The thawed samples were centrifuged at 15,000 rpm at 4°C for 30 min.

The pellets were washed with 70% ethanol and centrifuged at 15,000 rpm at 4°C for 5 min. The dried pellets were resuspended in 10 µl gel-loading dye (80% formamide, 10 mM EDTA, 0.1% xylene cyanol and 0.1% bromophenol blue) and incubated at 95°C for 5 min. Aliquots were analyzed on a 8% denaturing polyacrylamide/7 M urea PAGE using TBE buffer (89 mM Tris-borate (pH 8.3) and 1 mM EDTA) for 2 hr at 40 W. The gels were dried, exposed to a phosphorimager screen, and scanned using a Fuji FLA 9000 phosphor imager.

## Endonuclease assay

The assay was carried out as previously described (*Shibata et al., 2014*) with slight modifications. Briefly, the reaction mixtures (10 µl) without ScRev7 contained 100 ng of M13 circular ssDNA, 30 mM Tris–HCl, pH 7.5, 25 mM KCl, 5% glycerol, 1 mM DTT, 200 µg/ml BSA, 5 mM $MnCl_2$ and increasing concentrations of ScMre11 (0.1–2 µM). After incubation at 37°C for 1 hr, the reaction was stopped by adding 2 µl of stop solution (2 mg/ml proteinase K, 50 mM EDTA, and 1% SDS) and incubation was extended for 30 min, followed by the addition of 2 µl of gel loading solution. The reaction products were separated by electrophoresis on a 0.8% native agarose gel using 44.5 mM Tris-borate buffer (pH 8.3) containing 0.5 mM EDTA at 10 V/ cm for 1 hr. The gel was stained with ethidium bromide and the image was captured using the UVItec gel documentation system (UVItec, Cambridge, UK). To test the effect of ScRev7 on Mre11 endonuclease activity, the assay was carried out as described above, except that the ScRev7–Mre11 heterodimer was first formed by incubating a fixed amount of ScRev7 with increasing concentration of ScRev7 on ice for 15 min, prior to the addition of 10 µl of reaction buffer. The reaction mixtures were incubated and analyzed as described above.

## ATP crosslinking assay

The assay was carried out as previously described (*Thakur et al., 2021b*). The reaction mixture (20 µl) contained 50 mM Tris–HCl (pH 5.0), 10 mM $MgCl_2$, 400 pmol [γ-$^{32}$P]ATP and increasing concentrations of Rad50 (0.1–1 µM). Analogously, a fixed concentration of Rad50 (0.2 µM) was incubated with increasing concentrations of ScRev7 (0.5–6 µM) at 4°C for 15 min, prior to transferring the sample into 20 µl buffer containing 50 mM Tris–HCl (pH 5.0), 10 mM $MgCl_2$ and 400 pmol [γ-$^{32}$P]ATP. After incubation at 4°C for 25 min, samples were exposed to UV irradiation ($1.2 \times 10^5$ µJ/cm$^2$ in Hoefer UVC 500 ultraviolet crosslinker) at a distance of 2 cm. The reactions were stopped by adding 5 µl of 5× Laemmli buffer (10 mM Tris–HCl, pH 6.8, 12.5% SDS, 40% glycerol, and 0.1% bromophenol blue). Samples were incubated at 95°C for 10 min and resolved by SDS/PAGE in 10% polyacrylamide gel at 35 mA for 2 hr. The gels were dried, exposed to a phosphorimager screens, and scanned using a Fuji FLA-9000 phosphor imager and the band intensities of radiolabeled species was quantified using UVI-Band Map software (v. 97.04). Data were plotted as mean and SD in GraphPad Prism (v5.0).

## ATPase assay

The hydrolysis of [γ-$^{32}$P]ATP by Rad50 was assessed by measuring the release of $^{32}$Pi as previously described (*Thakur et al., 2021a*). A fixed concentration of Rad50 (0.5 µM) was incubated on ice for 30 min with increasing concentrations of ScRev7 or ScRev7-C1 to allow protein complex formation, prior to the reaction. Analogously, 400 pmol [γ-$^{32}$P] ATP and Rad50 (0.5 µM) were incubated in 10 µl reaction mixture containing 20 mM Tris–HCl (pH 7.5), 50 mM KCl, 0.2 mg/ml BSA, 0.1 mM DTT, 1.0 mM $MgCl_2$ and 5% glycerol, with indicated concentrations of Rad50. The reaction mixtures were incubated at 30°C for 30 min and the reaction was stopped by adding 15 mM EDTA. Aliquots (2 µl) from each sample were spotted onto a polyethyleneimine-cellulose plate, developed in a solution containing 0.5 M LiCl, 1 M formic acid and 1 mM EDTA. The reaction products were visualized by using a Fuji FLA-9000 phosphor imager, and the band intensities were quantified and plotted using UVI-Band map software. Data were plotted as mean and SD using GraphPad Prism (Version 5.0).

## Malachite green phosphate assay

ATP hydrolysis was monitored by measuring the amount of inorganic phosphate released using acidic ammonium molybdate and malachite green assay (*Lanzetta et al., 1979*). The reaction mixtures (80 µl) contained 20 mM Tris–HCl (pH 7.5), 50 mM KCl, 5% glycerol, 0.1 mM DTT, 0.2 mg/ml BSA, 1 mM $MgCl_2$, 150 µM ATP, and increasing concentrations of ScRad50 (0.05–1 µM). After incubation at 30°C for 30 min, reaction was stopped by adding 20 µl of malachite green reagent (Sigma-Aldrich,

MAK307). Incubation was continued for 15 min at 24°C to allow the formation of phosphomolybdate malachite green chromogenic complex. The absorbance values, as a measure of the extent of ATP hydrolysis, at 620 nm (y-axis) were plotted against increasing concentrations of ScRad50 (x-axis). To test the effect of ScRev7 or ScRev7-C1 on Rad50 ATPase activity, increasing concentrations of ScRev7 or ScRev7-C1 (0.1–2 µM) were incubated with 0.25 µM of ScRad50 on ice for 20 min, prior to mixing it with the reaction mixture. After incubation at 30°C for 30 min, ATP hydrolysis was monitored as described above. The data were plotted, and the best-fit line was determined by non-linear regression incorporating using GraphPad Prism (v. 5.0).

## Mass spectrometry analysis

Mass spectrometry analysis of Rev7-GFP was carried out to ascertain its identity using orbitrap mass spectrometry (*Zubarev and Makarov, 2013*). In-gel trypsin digestion was performed as follows: 50 mM ammonium bicarbonate and acetonitrile in 7:3 ratio was used for de-staining; following which, samples were reduced for 30 min by adding 10 mM DTT diluted in 50 mM ammonium bicarbonate. Following reduction, alkylation buffer (55 mM iodoacetamide in 50 mM ammonium bicarbonate) was added and samples were incubated at room temperature for 30 min. Samples were treated with 10 ng trypsin (Mass spectrometry-grade, Sigma-Aldrich, India), at 37°C for 12 hr. The peptides were eluted with 70% acetonitrile (300 µl) containing 0.1% trifluoracetic acid, and the samples were dried in speed vacuum and the pellet was resuspended in 40 µl resuspension buffer (2% acetonitrile in LC–MS grade Milli-Q). Samples (10 µl) were injected into the Orbitrap Fusion tribrid mass spectrometer (Thermo Fisher Scientific Inc, USA). Samples were run for 110 min using a nano-spray ionization source and static spray voltage. Ions were detected by Orbitrap detector at 60,000 resolution using quadrupole isolation. Daughter ions were detected by Ion Trap detector using quadrupole isolation mode. Following data acquisition, peptide spectrum matches and percentage peptide coverage were obtained for the samples. Collectively, these results confirmed that the purified protein is GFP-tagged ScRev7.

## NHEJ assay

The assay was performed using a linear plasmid as previously described (*Zhang and Paull, 2005*; *Ghodke and Muniyappa, 2013*). Briefly, 60 ng of BamHI-digested or undigested plasmid pRS416 was transformed into the WT and isogenic *rev1Δ*, *rev3Δ*, *rev7Δ*, *mre11Δ*, *sae2Δ*, *rev1Δrev3Δ*, *rev1Δrev7Δ*, *rev3Δrev7Δ*, *mre11Δrev7Δ*, *sae2Δrev7Δ rev7-C1*, or *rev7-42* strains. The uncut plasmid pRS416 served as a control. The transformants arising from plasmid re-circularization were selected on SC/-Ura agar plates after incubation at 30°C for 3–5 days. The efficiency of transformation was calculated as a ratio of the number of transformants with digested plasmid DNA to that with undigested plasmid DNA. The graph was obtained using GraphPad Prism (Version 5.0) and statistical significance was calculated using one-way ANOVA Dunnett's multiple comparisons test.

## Cell cycle analysis

Cell cycle analysis was performed using an FACS as previously described (*Ghodke and Muniyappa, 2016*). Cells were cultured to $A_{600}$ of 0.5 and synchronized in the G1 phase by treatment with α-factor (100 ng/ml, Sigma-Aldrich) for 2 hr at 30°C. The cells were washed three times with 1× phosphate-buffered saline and resuspended in fresh liquid YPD medium. Aliquots were collected after release from G1 arrest at 15, 30, 45, 60, 75, 90, 105, and 120 min. The cells were fixed in 70% ethanol and stored at 4°C. Subsequently, samples were treated with RNase A (0.4 mg/ml) at 37°C for 12 hr, stained with propidium iodide (15 µg/ml) and incubated at 37°C for an additional 3 hr. FACS analysis was performed on a FACSVerse analyzer (BD Biosciences). Dead and aggregated cells were excluded through gating, and cell cycle distribution was analyzed using FlowJo software (Version 10).

## NHEJ using a 'suicide-deletion' reporter assay

The assay was performed as previously described (*Karathanasis and Wilson, 2002*). Briefly, *S. cerevisiae* YW714 strains WT, *rev1Δ*, *rev3Δ*, *rev7Δ*, *mre11Δ*, *ku70Δ*, *rev7-C1*, and *rev7-42* cells were grown in an orbital shaking incubator at 200 rpm in a liquid SC/-Ura medium overnight at 30°C. The cells were pelleted by centrifugation at 4000 rpm for 10 min. Cell pellets were washed thrice with MilliQ water and resuspended in liquid SC medium at an $OD_{600}$ of 1. Serial dilutions were plated on SC/-Ura agar

plates containing glucose, and SC/-Ade agar plates containing galactose. After 5 days of incubation at 30°C, the number of colonies was counted. The rate of NHEJ, as a function of Ade[+] colonies, was calculated using the FALCOR software (*Hall et al., 2009*). The differences between experimental results and relevant controls were tested with Dunnett's multiple comparison test. The graph was generated using GraphPad Prism (Version 5.0). Statistical tests and p-values are mentioned in each figure legend.

## Plasmid-chromosome recombination assay

The assay was performed as previously described (*Paeschke et al., 2013*). The WT and isogenic *ura3-1*, *ura3-1 rev7Δ*, *ura3-1 mre11-D56N,H125N ura3-1 rev7Δ mre11-D56N,H125N*, *ura3-1 rev7-C1*, and *ura3-1 rev7-42* strains were transformed with 60 ng of empty vector (pFAT10) or a plasmid bearing G4 forming motifs (pFAT10-G4). The synchronous cells were grown in liquid SC/-Leu medium to an OD$_{600}$ of 0.5. Equal numbers of cells were replica-spotted on SC/-Ura medium and SC/-Ura medium plates containing 0.1 M HU. After incubation for 4–6 days at 30°C, the number of papillae was counted. The recombination frequency was calculated using FLCOR software (*Hall et al., 2009*).

## Quantitative PCR analysis

The rate of DNA end resection was determined in *rad51Δ*, *rad51Δ rev7Δ*, *rad51Δ mre11-H125N*, and *rad51Δ mre11-H125N rev7Δ* strains, as previously described (*Mimitou and Symington, 2010*; *Ferrari et al., 2018*). Single colonies grown in liquid YPD medium at 30°C for 12 hr were sub-cultured in liquid YP medium containing 2% raffinose at 30°C till $A_{600}$ reached 0.5. At this stage, cells were synchronized at G2/M phase by adding nocodazole (10 μg/ml) and HO expression was induced by adding 2% galactose to the cell culture. For each time point, 20 ml samples were collected for each strain and 0.1% sodium azide was added immediately. Cells were collected by centrifugation at 4000 rpm for 10 min. The bead-beating step was used for isolation of genomic DNA from the resuspended pellets (*Amberg et al., 2005*). Samples were incubated with 10 units of StyI-HF and XbaI-HF (New England Biolabs, Ipswich, USA) in a reaction mixture containing 15 μg of genomic DNA (gDNA), 1× CutSmart buffer at 37°C for 7 hr. In parallel, equal amounts of gDNA was mock-digested as control. DNA was precipitated by adding equal volumes of isopropanol, followed by centrifugation at 13,000 rpm for 20 min. Pellets were washed with 1 ml of 70% ethanol, air dried, and resuspended in 50 μl of 1× Tris–EDTA buffer (pH 7.5). In qPCR analysis, equal amounts of DNA (1 ng) were used per reaction in a mixture containing 1× SYBR green master mix (G-Biosciences, St. Louis, MO) and 0.2 μM each of forward and reverse primers, as indicated in *Supplementary file 1h*. The qPCR reaction was performed using a Bio-Rad CFX96 thermocycler and 96-well PCR plates. The data obtained from the qPCR were analyzed using the Bio-Rad CFX Maestro 1.1 software (version 2.3). The specificity of the qPCR was ascertained by (1) agarose gel electrophoresis and (2) melting curve analysis of qPCR products. Similarly, primer efficiencies were calculated by performing qPCR analysis using serially diluted DNA samples. The HO cut efficiencies across strains were estimated by qPCR analysis at a single HO site (Coordinates 284324–294420) with a unique upstream sequence on Chromosome III. Upon DSB induction, the percentage of ssDNA generated by end resection at different time points and distance (0.7 and 3 kb) from the break was calculated as previously described (*Ferrari et al., 2018*). The *PRE1* gene situated on chromosome V was used as an internal reference. Two-way ANOVA was performed to statistically analyse the datasets and graphs were generated using GraphPad Prism (Version 5.0).

## Statistical and data analysis

Differences among groups were analyzed by one-way ANOVA Dunnett's multiple comparison test followed by Tukey's post hoc test (GraphPad Prism 6.07). Statistical significance level was set as follows: *p-value <0.05; **p-value <0.01; ***p-value <0.001; ****p-value <0.0001. The statistical methods employed to analyse the data are indicated in the figures legends.

## Acknowledgements

We thank Drs. Virginia Zakian, Maria Pia Longhese, Lorraine Symington, and Thomas Wilson for kindly providing some of the strains used in this study, as well as to Dr. Narottam Acharya for the generous gift of ScRev7 and ScRev1 expression plasmids, *S. cerevisiae* BJ5464 strain, and for his assistance in the purification of expressed proteins, and Naren Chandran Shakthivel for his assistance with the

generation of AlphaFold-multimer models. This work was supported by a grant (CRG/2021/000082) from the Science and Engineering Research Board, New Delhi to KM, who was also the recipient of Bhatnagar Fellowship (SP/CSIR/425/2018) from the Council of Scientific and Industrial Research, New Delhi.

## Additional information

### Funding

| Funder | Grant reference number | Author |
|---|---|---|
| Science and Engineering Research Board | CRG/2021/000082 | Kalappa Muniyappa |
| Council of Scientific and Industrial Research, India | SP/CSIR/425/2018 | Kalappa Muniyappa |

The funders had no role in study design, data collection, and interpretation, or the decision to submit the work for publication.

### Author contributions

Sugith Badugu, Software, Validation, Investigation, Methodology, Writing – original draft; Kshitiza Mohan Dhyani, Manoj Thakur, Data curation, Software, Validation, Investigation, Methodology, Writing – original draft; Kalappa Muniyappa, Conceptualization, Resources, Formal analysis, Supervision, Funding acquisition, Project administration, Writing – review and editing

### Author ORCIDs

Kshitiza Mohan Dhyani ![ORCID] https://orcid.org/0000-0002-1234-202X
Manoj Thakur ![ORCID] https://orcid.org/0000-0001-5347-3344
Kalappa Muniyappa ![ORCID] https://orcid.org/0000-0002-9192-9194

Reviewer #1 (Public review): https://doi.org/10.7554/eLife.96933.3.sa1
Reviewer #2 (Public review): https://doi.org/10.7554/eLife.96933.3.sa2
Reviewer #3 (Public review): https://doi.org/10.7554/eLife.96933.3.sa3
Author response https://doi.org/10.7554/eLife.96933.3.sa4

## Additional files

### Supplementary files

• Supplementary file 1. Inter-atomic distances and confidence parameters of amino acid residues mediating Rev7–Mre11 interactions. Residues in bold are present in the C-terminal safety-belt region of Rev7 protein.

• Supplementary file 2. Inter-atomic distances and confidence parameters of amino acid residues mediating Rev7–Rad50 interactions. Residues in bold are present in the C-terminal safety-belt region of Rev7 protein.

• Supplementary file 3. The exact p-values for *Figure 7B*. The p-values were obtained by comparing the percentage of non-homologous end-joining observed for the indicated single- or double-gene deletions versus either the wild-type (WT) or *rev7Δ* strain, using non-parametric one-way ANOVA Dunnett test.

• Supplementary file 4. The exact p-values for *Figure 8B*. The p-values were obtained by comparing the percentage of non-homologous end-joining observed for the indicated single- or double-gene deletions versus either the wild-type (WT) or *rev7Δ* strain, using non-parametric one-way ANOVA Dunnett test.

• Supplementary file 5. *S. cerevisiae* strains used in this study.

• Supplementary file 6. Sequences of primers used in this study. The bold letters correspond to restrictions sites.

• Supplementary file 7. Sequences of oligonucleotides used in the preparation of DNA substrates.

- Supplementary file 8. Primers used for qPCR analysis.
- MDAR checklist

### Data availability

All data generated or analyzed during this study are included in the manuscript and supplementary information. Source data files have been provided for Figure 1—figure supplement 3, Figure 1—figure supplement 4, Figure 2—figure supplement 2, Figure 5, Figure 5—figure supplement 1, Figure 6, Figure 7, Figure 8, and Figure 9—figure supplement 2.

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
