## [Editor Report · eLife Assessment]

This manuscript reports **important** data providing evidence that a 42 amino acid region of Rev7 is necessary and sufficient for interaction with the Rad50-Mre11-Xrs2 complex in budding yeast. The authors conclude that Rev7 inhibits the Rad50 ATPase and the Mre11 nuclease with the exception of ssDNA exonuclease activity. The **convincing** data largely support the conclusions, although the effect of Rev7 on homologous recombination is less well documented and the observed effect on resection is moderate. Specifically, the result that the Rev7 C-terminal truncation lacking the 42 amino acid region still suppresses homologous recombination is unexpected and unexplained.

---

## [Referee Report · Reviewer #1 (Public review)]

Summary:

The mammalian Shieldin complex consisting of REV7 (aka MAD2L2, MAD2B) and SHLD1-3 affects pathway usage in DSB repair favoring non-homologous endjoining (NHEJ) at the expense of homologous recombination (HR) by blocking resection and/or priming fill-in DNA synthesis to maintain or generate near blunt ends suitable for NHEJ. While the budding yeast *Saccharomyces cerevisiae* does not have homologs to SHLD1-3, it does have Rev7, which was identified to function in conjunction with Rev3 in the translesion DNA polymerase zeta. Testing the hypothesis that Rev7 also affect DSB resection in budding yeast, the work identified a direct interaction between Rev7 and the Rad50-Mre11-Xrs2 complex by two-hybrid and direct protein interaction experiments. Deletion analysis identified that the 42 amino acid C-terminal region was necessary and sufficient for the 2-hybrid interaction. Direct biochemical analysis of the 42 aa peptide was not possible. Rev7 deficient cells were found to be sensitive to HU only in synergy with G2 tetraplex forming DNA. Importantly, the 42 aa peptide alone suppressed this phenotype. Biochemical analysis with full-length Rev7 and a C-terminal truncation lacking the 42 aa region shows G4-specific DNA binding that is abolished in the C-terminal truncation and with a substrate containing mutations to prevent G4 formation. Rev7 lacks nuclease activity but inhibits the dsDNA exonuclease activity of Mre11. The C-terminal truncation protein lacking the 42 aa region also showed some inhibition suggesting the involvement of additional binding sites besides the 42 aa region. Also, the Mre11 ssDNA endonuclease activity is inhibited by Rev7 but not the degradation of linear ssDNA. Rev7 does not affect ATP binding by Rad50 but inhibits in a concentration-dependent manner the Rad50 ATPase activity. The C-terminal truncation protein lacking the 42 aa region also showed some inhibition but significantly less than the full-length protein. Using an established plasmid-based NHEJ assay, the authors provide strong evidence that Rev7 affects NEHJ, showing a four-fold reduction in this assay. The mutations in the other Pol zeta subunits, Rev3 and Rev1, show a significantly smaller effect (~25% reduction). A strain expressing only the Rev7 C-terminal 42 aa peptide showed no NHEJ defect, while the truncation protein lacking this region exhibited a smaller defect than the deletion of REV7. The conclusion that Rev7 supports NHEJ mainly through the 42 aa region was validated using a chromosomal NHEJ assay. The effect on HR was assessed using a plasmid:chromosome system containing G4 forming DNA. The rev7 deletion strain showed an increase in HR in this system in the presence and absence of HU. Cells expressing the 42 aa peptide were indistinguishable from wild type as were cells expressing the Rev7 truncation lacking the 42 aa region. The authors conclude that Rev7 suppresses HR, but the context appears to be system-specific and the conclusion that Rev7 abolished HR repair of DSBs is unwarranted and overly broad.

Strength:

This is a well-written manuscript with well-executed experiments which suggest that Rev7 inhibits MRX-mediated resection to favor NEHJ during DSB repair. This finding is novel and provides insight into the potential mechanism of how the human Shieldin complex might antagonize resection.

Weaknesses:

The nuclease experiments were conducted using manganese as a divalent cation, and it is unclear whether there is an effect with the more physiological magnesium cation. The data largely support the conclusions, although the effect of Rev7 on HR is less well documented, as only a highly specialized assay is used that does not warrant the broad conclusion drawn. Specifically, the results that the Rev7 c-terminal truncation lacking the 42 aa region still suppresses HR is unexpected and unexplained.

In this revision the authors addressed most of my concerns by text revisions and addition of new data.

The new two hybrid data showing that the 42 amino acid segment interacts with MRN are valuable. However, it may not be clear to which subunit the 42 aa segment binds, as in the yeast 2H system the chromosomally encoded subunits are present or were the 2H experiments conducted in an MRN deletion background?. This could be acknowledged.

The material and methods section was updated to indicate use of 5 mM MnCl2 and 5 mM MgCl2 in the exonuclease assay but not the endonuclease assay. Please check if this is correct. Why the difference between both assays? There is a concern that the absence of ATP and Mg affects the endonuclease assay.

The addition of Dmc1 as a specificity control for the ATPase inhibition is nice and shows a specific effect. The use of Sae2 associated nuclease activity as a specificity control for the nuclease inhibition is problematic. There has been considerable debate about the Sae2 associated nuclease activity, which seems to have been solved by the Cejka lab showing that Sae2 is a cofactor of MRN without intrinsic nuclease activity (e.g. https://pubmed.ncbi.nlm.nih.gov/25231868/). Or do the authors want to suggest that Sae2 has intrinsic nuclease activity? The control may still be useful mentioning that the nuclease is associated but not intrinsic and citing the relevant papers.

---

## [Referee Report · Reviewer #2 (Public review)]

In this study, Badugu et al investigate the Rev7 roles in regulating the Mre11-Rad50-Xrs2 complex and in metabolism of G4 structures. The authors also try to make a conclusion that REV7 can regulate the DSB repair choice between homologous recombination and non-homologous end joining.

The major observations of this study are:

(1) Rev7 interacts with the individual components of the MRX complex in a two-hybrid assay and in a protein-protein interaction assay (microscale thermophoresisi) in vitro.

(2) Modeling using AlphaFold-Multimier also indicated that Rev7 can interact with Mre11 and Rad50.

(3) Using a two-hybrid assay, a 42 C terminal domain in Rev7 responsible for the interaction with MRX was identified.

(4) Rev7 inhibits Mre11 nuclease and Rad50 ATPase activities in vitro.

(5) Rev 7 promotes NHEJ in plasmid cutting/relegation assay.

(6) Rev7 inhibits recombination between chromosomal ura3-1 allele and plasmid ura3 allele containing G4 structure.

(7) Using an assay developed in V. Zakian's lab, it was found that rev7 mutants grow poorly when both G4 is present in the genome and yeast are treated with HU.

(8) In vitro, purified Rev7 binds to G4-containing substrates.

In general, a lot of experiments have been conducted, but the major conclusion about the role of Rev7 in regulating the choice between HR and NHEJ is not justified.

(1) Two stories that do not overlap (regulation of MRX by Rev7 and Rev7 role in G4 metabolism) are brought under one umbrella in this work. There is no connection unless the authors demonstrate that Rev7 inhibits the cleavage of G4 structures by the MRX complex.

(2) The authors cannot conclude based on the recombination assay between G4-containing 2-micron plasmid and chromosomal ura3-1 that Rev7" completely abolishes DSB-induced HR". First of all, there is no evidence that DSBs are formed at G4. Why is there no induction of recombination when cells are treated with HU? Second, as the authors showed, Rev7 binds to G4, therefore it is not clear if the observed effects are the result of Rev7 interaction with G4 or impact on HR. The established HO-based assays where the speed of resection can be monitored (e.g., Mimitou and Symington, 2010) have to be used to justify the conclusion that Rev7 inhibits MRX nuclease activity in vivo.

Comments on the revised version:

I am satisfied with the revision. Specifically, (i) the elimination of the G4 part and (ii) the implementation of the HO-endonuclease resection assay described in Mimiou and Symington, 2010 significantly improved the clarity of the work and strengthened the conclusion about the Rev7 interference with DNA resection.

---

## [Referee Report · Reviewer #3 (Public review)]

Summary:

REV7 facilitates the recruitment of Shieldin complex and thereby inhibits end resection and controls DSB repair choice in metazoan cells. Puzzlingly, Shieldin is absent in many organisms, and it is unknown if and how Rev7 regulates DSB repair in these cells. The authors surmised that yeast Rev7 physically interacts with Mre11/Rad50/Xrs2 (MRX), the short-range resection nuclease complex and tested this premise using yeast two hybrid (Y2H) and microscale thermophoresis (MST). The results convincingly showed that the individual subunits of MRX interacts robustly with Rev7. By AlphaFold Multimer modelling followed by Y2H confirmed that the carboxy terminal 42 amino acid is essential for interaction with MR and G4 DNA binding by REV7. The mutant rev7 lacking the binding interface (Rev7-C1) to MR shows moderate inhibition to the nuclease and the ATPase activity of Mre11/Rad50 in biochemical assays. Deletion of REV7 also causes a mild reduction in NHEJ using both plasmid and chromosome-based assays and increases mitotic recombination between chromosomal ura3-01 and the plasmid ura3 allele interrupted by G4. The revision also showed that rev7 deleted cells exhibit mild hyper-resection phenotype at 0.7 and 3 kb from the DSB using qPCR assays. The authors concluded that Rev7 facilitates NHEJ and antagonises HR even in budding yeast, but it achieves this by blocking Mre11 nuclease and Rad50 ATPase.

Weaknesses:

There are several strengths to the studies and the broad types of well-established assays were used to deduce the conclusion. Nevertheless, there are notable discrepancies on the mutant phenotypes that were to test the functionality of Rev7-MRX interaction on the repair outcomes, raising concerns on the validity of the proposed model. The manuscript also needs a few additional functional assays to reach the accurate conclusions as proposed. The revision responded to several comments raised by the reviewers, but they are inadequate to address the key concerns and did not offer sufficient and compelling experimental support to the main premise that Rev7-Mre11/Rad50/Xrs2 interactions regulate MRX activities in cells and thereby modulates DSB repair choice in budding yeast.

(1) AlphaFold model predicts that Mre11-Rev7 and Rad50-Rev7 binding interfaces overlap and Rev7 might bind only to Mre11 or Rad50 at a time. Interestingly, however, Rev7 appears dimerized (Fig.1). Since MR complex also forms with 2M and 2R in the complex, it should still be possible if REV7 can interact both M and R in the MR complex. The author should perform MST using MR complex instead of individual MR components. The authors should also analyze if Rev7-C1 is indeed deficient in interaction with MR individually and with complex using MST assay.

(2) The nuclease and the ATPase assays require additional controls. Does Rev7 inhibit the other nuclease or ATPase non-specifically? Are these outcomes due to the non-specific or promiscuous activity of Rev7? In fig.6, the effect of REV7 on the ATP binding of Rad50 could be hard to assess because the maximum Rad50 level (1 uM) was used in the experiments. The author should use the suboptimal level of Rad50 to check if REV7 still does not influence ATP binding by Rad50.

(3) The moderate deficiency in NHEJ using plasmid based assay in REV7 deleted cells can be attributed to aberrant cell cycle or mating type in rev7 deleted cells. The authors should demonstrate that rev7 deleted cells retain largely normal cell cycle pattern and the mating type phenotypes. The author should also analyze the breakpoints in plasmid based NHEJ assays in all mutants especially from rev7 and rev7-C1 cells.

(4) It is puzzling why the authors did not analyze end resection defects in rev7 deleted cells after a DSB. The author should employ the widely used resection assay after a HO break in rev3, rev7 and mre11 rev7 cells as described previously.

(5) Is it possible that Rev7 also contributes to NHEJ as the part of TLS polymerase complex? Although NHEJ largely depends on Pol4, the authors should not rule out the possibility if the observed NHEJ defect in rev7 cells are due at least partially to its well-known TLS defect and not all due to their role in MRX activity regulation as the authors proposed. In fact, rev3 or rev1 cells are partially defective in NHEJ (Fig. 7). Rev7-C1 is less deficient in NHEJ than REV7 deletion. These results predict that rev7-C1 rev3 could be more deficient than rev3 or rev7-C1, and such results might indicate that Rev7 contributes to NHEJ by two ways; one by interacting (and modulating) MRX and the other as part of Rev3-Rev7 complex. Additionally, the authors should examine if Rev7-C1 might be deficient in TLS. In this regard, does rev7-C1 reduce TLS and TLS dependent mutagenesis? Is it dominant? The authors should also check if Rev3/Rev1 complexes are stable in Rev7 deleted or rev7-C1 cells by immunoblot assays.

(6) Due to the G4 DNA and G4 binding activity of REV7, it is not clear which class of events the authors are measuring in plasmid-chromosome recombination assay in Fig.9. Do they measure G4 instability or the integrity of recombination or both in rev7 deleted cells. Instead, the effect of rev7 deletion or rev7-C1 on recombination should be measured directly by more standard mitotic recombination assays like mating type switch or his3 repeat recombination. The revision did not address these concerns, which still makes the interpretation of the provided recombination results difficult.

---

## [Author Response]

The following is the authors’ response to the original reviews.

**Public Reviews:**

**Reviewer 1 (Public Review):**
SummaryThe mammalian Shieldin complex consisting of REV7 (aka MAD2L2, MAD2B) and SHLD1-3 affects pathway usage in DSB repair favoring non-homologous end-joining (NHEJ) at the expense of homologous recombination (HR) by blocking resection and/or priming fill-in DNA synthesis to maintain or generate near blunt ends suitable for NHEJ. While the budding yeast *Saccharomyces cerevisiae* does not have homologs to SHLD1-3, it does have Rev7, which was identified to function in conjunction with Rev3 in the translesion DNA polymerase zeta. Testing the hypothesis that Rev7 also affects DSB resection in budding yeast, the work identified a direct interaction between Rev7 and the Rad50-Mre11-Xrs2 complex by two-hybrid and direct protein interaction experiments. Deletion analysis identified that the 42 amino acid C-terminal region was necessary and sufficient for the 2-hybrid interaction. Direct biochemical analysis of the 42 aa peptide was not possible. Rev7 deficient cells were found to be sensitive to HU only in synergy with G2 tetraplex forming DNA. Importantly, the 42 aa peptide alone suppressed this phenotype. Biochemical analysis with full-length Rev7 and a C-terminal truncation lacking the 42 aa region shows G4-specific DNA binding that is abolished in the C-terminal truncation and with a substrate containing mutations to prevent G4 formation. Rev7 lacks nuclease activity but inhibits the dsDNA exonuclease activity of Mre11. The C-terminal truncation protein lacking the 42 aa region also showed some inhibition suggesting the involvement of additional binding sites besides the 42 aa region. Also, the Mre11 ssDNA endonuclease activity is inhibited by Rev7 but not the degradation of linear ssDNA. Rev7 does not affect ATP binding by Rad50 but inhibits in a concentration-dependent manner the Rad50 ATPase activity. The C-terminal truncation protein lacking the 42 aa region also showed some inhibition but significantly less than the full-length protein.Using an established plasmid-based NHEJ assay, the authors provide strong evidence that Rev7 affects NEHJ, showing a four-fold reduction in this assay. The mutations in the other Pol zeta subunits, Rev3 and Rev1, show a significantly smaller effect (~25% reduction). A strain expressing only the Rev7 C-terminal 42 aa peptide showed no NHEJ defect, while the truncation protein lacking this region exhibited a smaller defect than the deletion of REV7. The conclusion that Rev7 supports NHEJ mainly through the 42 aa region was validated using a chromosomal NHEJ assay. The effect on HR was assessed using a plasmid:chromosome system containing G4 forming DNA. The rev7 deletion strain showed an increase in HR in this system in the presence and absence of HU. Cells expressing the 42 aa peptide were indistinguishable from the wild type as were cells expressing the Rev7 truncation lacking the 42 aa region. The authors conclude that Rev7 suppresses HR, but the context appears to be system-specific and the conclusion that Rev7 abolished HR repair of DSBs is unwarranted and overly broad.StrengthThis is a well-written manuscript with many well-executed experiments that suggest that Rev7 inhibits MRX-mediated resection to favor NEHJ during DSB repair. This finding is novel and provides insight into the potential mechanism of how the human Shieldin complex might antagonize resection.

We thank Reviewer 1 for their comprehensive summary of our work. The Reviewers' recognition that our manuscript is “well-written” with “many well-executed experiments” and our findings are “novel” is greatly appreciated.

WeaknessesThe nuclease experiments were conducted using manganese as a divalent cation, and it is unclear whether there is an effect with the more physiological magnesium cation. Additional controls for the ATPase and nuclease experiments to eliminate non-specific effects would be helpful. Evidence for an effect on resection in cells is lacking. The major conclusion about the role of Rev7 in regulating the choice between HR and NHEJ is not justified, as only a highly specialized assay is used that does not warrant the broad conclusion drawn. Specifically, the results that the Rev7 C terminal truncation lacking the 42 aa region still suppresses HR is unexpected and unexplained. The effect of Rev7 on G4 metabolism is underdeveloped and distracts from the main results that Rev7 modulated MRX activity. The authors should consider removing this part and develop a more complete story on this later.

We have addressed each point identified as “Weaknesses” by the reviewer, as described below:

The nuclease experiments were conducted using manganese as a divalent cation, and it is unclear whether there is an effect with the more physiological magnesium cation.

We acknowledge the Reviewer’s concern and apologize for not having been clear in our first submission. However, several studies have demonstrated that Mre11 exhibits all three DNase activities, namely single-stranded endonuclease, double-stranded exonuclease and DNA hairpin opening only in the presence of Mn²⁺ but not with other divalent cations, such as magnesium or calcium (Paull and Gellert, Mol. Cell 1998; 2000; Usui et al., Cell 1998; Ghosal and Muniyappa, JMB, 2007; Arora et al., Mol Cell Biol. 2017). For this reason, Mn²⁺ was used as a cofactor for the Mre11 nuclease assays. We have clarified this in the revised manuscript. As a side note, Mg2+ serves as a cofactor for Rad50’s ATPase activity.

Additional controls for the ATPase and nuclease experiments to eliminate non-specific effects would be helpful.

We thank the Reviewer for raising this important point, as it led us to evaluate and confirm the specificity of Rev7 and exclude its potential non-specific effects. To this end, we have performed additional experiments, which showed that (a) the *S. cerevisiae* Dmc1 ATPase activity was not affected by Rev7, contrary to its inhibitory effect on Rad50 and (b) Rev7 had no discernible impact on the endonucleolytic activity of *S. cerevisiae* Sae2, whereas it inhibits DNase activities of Mre11. Thus, the lack of inhibitory effects on the ATPase activity of Dmc1 and nuclease activity of Sae2 confirm the specificity of Rev7 for Mre11 and Rad50 subunits. We have included this new data in Figure 6H and 6J and in Figure 5 –figure supplement 1, respectively, in the revised manuscript.

Evidence for an effect on resection in cells is lacking. The major conclusion about the role of Rev7 in regulating the choice between HR and NHEJ is not justified, as only a highly specialized assay is used that does not warrant the broad conclusion drawn.

We agree with the Reviewer that in vivo evidence demonstrating the inhibitory effect of REV7 on DNA end resection was lacking in the first submission. Reviewer 2 and 3 have also raised point. We now measured the rate of DNA end resection using a qPCR-based assay (Mimitou and Symington, EMBO J. 2010; Gnugge et al., Mol. Cell 2023). The results revealed that deletion of REV7 led to an enhancement in the rate of DNA end resection at a DSB site inflicted by HO endonuclease (Figure 9—figure supplement 3), providing direct evidence that loss of *REV7* contributes to increase in DNA end resection at the DSBs.

Specifically, the results that the Rev7 C-terminal truncation lacking the 42 aa region still suppresses HR is unexpected and unexplained.

This is a fair point, and we thank the reviewer for raising it. Although the interaction of Rev7-C1 in the yeast two-hybrid assays was not apparent, surprisingly, it partially suppressed HR (Figure 9). In line with this, biochemical assays showed that it exerts partial inhibitory effect on the Mre11 nuclease (Figure 5) and Rad50 ATPase (Figure 6) activities compared with the full-length Rev7. Consistent with *vitro* data, the AF2 models revealed that, in addition to the C-terminal 42-aa region, residues in the N-terminal region of Rev7 also interact with the Mre11 and Rad50 subunits (Figure 2—figure supplement 2).

The effect of Rev7 on G4 metabolism is underdeveloped and distracts from the main results that Rev7 modulated MRX activity. The authors should consider removing this part and develop a more complete story on this later.

We agree with the reviewer’s comment “that the effect of Rev7 on G4 DNA metabolism is underdeveloped and distracts” from the central theme of the present paper, and suggested that we develop this part as a complete story later. This point has also been raised by Reviewer 2 and 3 and, therefore, Figures and associated text were removed in the revised version of the manuscript.

**Reviewer 2 (Public Review):**
In this study, Badugu et al investigate the Rev7 roles in regulating the Mre11-Rad50-Xrs2 complex and in the metabolism of G4 structures. The authors also try to make a conclusion that REV7 can regulate the DSB repair choice between homologous recombination and non-homologous end joining.The major observations of this study are:(1) Rev7 interacts with the individual components of the MRX complex in a two-hybrid assay and in a protein-protein interaction assay (microscale thermophoresisi) in vitro.(2) Modeling using AlphaFold-Multimier also indicated that Rev7 can interact with Mre11 and Rad50.(3) Using a two-hybrid assay, a 42 C terminal domain in Rev7 responsible for the interaction with MRX was identified.(4) Rev7 inhibits Mre11 nuclease and Rad50 ATPase activities in vitro.(5) Rev 7 promotes NHEJ in plasmid cutting/relegation assay.(6) Rev7 inhibits recombination between chromosomal ura3-1 allele and plasmid ura3 allele containing G4 structure.(7) Using an assay developed in V. Zakian's lab, it was found that rev7 mutants grow poorly when both G4 is present in the genome and yeast are treated with HU.(8) In vitro, purified Rev7 binds to G4-containing substrates.In general, a lot of experiments have been conducted, but the major conclusion about the role of Rev7 in regulating the choice between HR and NHEJ is not justified.

We appreciate Reviewer 2 for comprehensive assessment of our manuscript and their insightful comments. However, we believe that the data (Figure 7-9) in our manuscript, together with new data (Figure 9- figure supplement 2 and 3) in the revised manuscript, clearly demonstrate that Rev7 regulates the choice between HR and NHEJ.

(1) Two stories that do not overlap (regulation of MRX by Rev7 and Rev7's role in G4 metabolism) are brought under one umbrella in this work. There is no connection unless the authors demonstrate that Rev7 inhibits the cleavage of G4 structures by the MRX complex.

We agree with the reviewer’s point that the themes associated with the regulation of the functions of MRX subunits by Rev7 and its role G4 DNA metabolism do not overlap. This concern has also been expressed by Reviewer 1 and 3. According to their suggestion, we have deleted all figures and text describing the role of Rev7 in G4 DNA metabolism from the revised manuscript.

(2) The authors cannot conclude based on the recombination assay between G4-containing 2-micron plasmid and chromosomal ura3-1 that Rev7 "completely abolishes DSB-induced HR". First of all, there is no evidence that DSBs are formed at G4. Why is there no induction of recombination when cells are treated with HU? Second, as the authors showed, Rev7 binds to G4, therefore it is not clear if the observed effects are the result of Rev7 interaction with G4 or its impact on HR. The established HO-based assays where the speed of resection can be monitored (e.g., Mimitou and Symington, 2010) have to be used to justify the conclusion that Rev7 inhibits MRX nuclease activity in vivo.

We thank the Reviewer for the insightful comments and drawing our attention to the inference *"completely abolishes DSB-induced HR"*. We have we have rephrased the conclusion, and replaced it with *“REV7* gene product plays an anti-recombinogenic role during HR”. Then, the reviewer refers to lack of “*evidence that DSBs are formed at G4”.* At this point, unfortunately, our attempts to identify DSB at the G4 DNA site in the 2-micron plasmid did not provide a clear answer to this question. This might be related to the existence of myriad DNases in the cell and technical issues associated with the isolation of low-abundant, linearized 2-micron plasmid molecules. Because of these reasons, we cannot provide any data on DSB at the G4 site in the 2-micron plasmid.

The reviewer then correctly points out “Why is there no induction of recombination when cells are treated with HU?” These findings are consistent with previous studies which have shown that Mre11-deﬁcient cells are sensitivity to HU, resulting in cell death (Tittel-Elmer et al., EMBO J. 28, 1142-1156, 2009; Hamilton and Maizels, PLoS One, 5, e15387, 2010). However, a novel finding of our study is that ura3-1 rev7D cells and ura3-1 cells expressing Rev7-42 amino acid peptide (to limited extent) produce Ura3+ papillae. We have included this information in the Results section and adjusted the text to make this point clear to the reader.

In the same paragraph, the Reviewer expresses a concern about the interaction of Rev7 with G4 DNA substrates and its impact on HR. As discussed above, in response to your comment (1) and a similar comment of Reviewer 1 and 3, we have deleted all figures and text describing the role of Rev7 in G4 DNA metabolism in the revised manuscript. The reviewer specifically refers to a study by Mimitou and Symington, 2010 in which the speed DNA end resection at the HO endonuclease-inflicted DSB was quantified. We have carried out the suggested experiment and the results are presented in Figure 9─figure supplement 3.

**Reviewer 3 (Public Review):**
Summary:REV7 facilitates the recruitment of Shieldin complex and thereby inhibits end resection and controls DSB repair choice in metazoan cells. Puzzlingly, Shieldin is absent in many organisms and it is unknown if and how Rev7 regulates DSB repair in these cells. The authors surmised that yeast Rev7 physically interacts with Mre11/Rad50/Xrs2 (MRX), the short-range resection nuclease complex, and tested this premise using yeast two-hybrid (Y2H) and microscale thermophoresis (MST). The results convincingly showed that the individual subunits of MRX interact robustly with Rev7. AlphaFold Multimer modelling followed by Y2H confirmed that the carboxy-terminal 42 amino acid is essential for interaction with MR and G4 DNA binding by REV7. The mutant rev7 lacking the binding interface (Rev7-C1) to MR shows moderate inhibition to the nuclease and the ATPase activity of Mre11/Rad50 in biochemical assays. Deletion of REV7 also causes a mild reduction in NHEJ using both plasmid and chromosome-based assays and increases mitotic recombination between chromosomal ura3-01 and the plasmid ura3 allele interrupted by G4. The authors concluded that Rev7 facilitates NHEJ and antagonizes HR even in budding yeast, but it achieves this by blocking Mre11 nuclease and Rad50 ATPase.WeaknessesThere are many strengths to the studies and the broad types of well-established assays were used to deduce the conclusion. Nevertheless, I have several concerns about the validity of experimental settings due to the lack of several key controls essential to interpret the experimental results. The manuscript also needs a few additional functional assays to reach the accurate conclusions as proposed.

We are happy that the Reviewer has found “many strengths” in our manuscript and further noted that “results convincingly showed that the individual subunits of MRX interact robustly with Rev7”. We greatly appreciate the Reviewer for these encouraging words, and for specific suggestions that helped us to improve the manuscript. As suggested, we have performed additional experiments including key controls and the data is presented in the revised manuscript.

(1) AlphaFold model predicts that Mre11-Rev7 and Rad50-Rev7 binding interfaces overlap and Rev7 might bind only to Mre11 or Rad50 at a time. Interestingly, however, Rev7 appears dimerized (Figure 1). Since the MR complex also forms with 2M and 2R in the complex, it should still be possible if REV7 can interact with both M and R in the MR complex. The author should perform MST using MR complex instead of individual MR components. The authors should also analyze if Rev7-C1 is indeed deficient in interaction with MR individually and with complex using MST assay.

Thank you for the valuable suggestion. As requested, MST titration experiments have been performed to examine the affinity of purified GFP-tagged Rev7-C1 for the Mre11, Rad50 and MR complex. The results revealed that Rev7-C1 binds to the Mre11 and Rad50 subunits with about 3- and 8.8-fold reduced affinity, respectively; whereas it binds to the MR complex with ~5.6-fold reduced affinity compared with full-length Rev7. The data is shown in Figure 1─figure supplement 4A-C.

(2) The nuclease and the ATPase assays require additional controls. Does Rev7 inhibit the other nuclease or ATPase non-specifically? Are these outcomes due to the non-specific or promiscuous activity of Rev7? In Figure 6, the effect of REV7 on the ATP binding of Rad50 could be hard to assess because the maximum Rad50 level (1 mM) was used in the experiments. The author should use the suboptimal level of Rad50 to check if REV7 still does not influence ATP binding by Rad50.

We thank the Reviewer for these valuable comments (Reviewer 1 has raised similar issues). Thus, we performed additional control experiments and the results indicate that (a) the ATPase activity of *S. cerevisiae* Dmc1 was not affected by Rev7 and (b) Rev7 does not inhibit the endonucleolytic activity of *S. cerevisiae* Sae2. The results are depicted in Figure 6H and 6J and Figure 5 –figure supplement 1A-D, respectively.

As suggested by the Reviewer, using suboptimal levels of Rad50 (0.2 mM), we carried out experiments to test the effect of varying concentrations of Rev7 on the ability of Rad50 to bind ATP and catalyse its hydrolysis. The results showed that Rev7 had no discernible effect on its ability to bind ATP, even at concentrations 30 times higher than the concentration of Rad50 (Figure 6B and 6D). However, Rev7 suppresses the ATPase activity of Rad50, but not that of Dmc1, in a concentration-dependent manner (Figure G, 6J).

(3) The moderate deficiency in NHEJ using plasmid-based assay in REV7 deleted cells can be attributed to aberrant cell cycle or mating type in rev7 deleted cells. The authors should demonstrate that rev7 deleted cells retain largely normal cell cycle patterns and the mating type phenotypes. The author should also analyze the breakpoints in plasmid-based NHEJ assays in all mutants, especially from rev7 and rev7-C1 cells.

We appreciate the Reviewer's critical and insightful comment. We monitored cell-cycle progression of both wild-type and *rev7D* cells over time using FACS. The results revealed that the cell cycle profiles and mating type phenotypes *rev7D* cells were similar to the wild type cells. The data is presented in Figure 7-figure supplement 1. This indicates that *rev7D* cells do not possess aberrant cell cycle or mating type defects as compared with the wild-type cells.

We find the second point raised by the Reviewer although is intriguing, its relevance to the current study is unclear. In our view, identification of breakpoints using plasmid-based NHEJ assays in all the mutants will require a significant amount of time, and the insight that we may gain is unlikely to add to the central theme of this paper. Moreover, we know for sure that Rev7 has no DNA cleavage/nicking activity.

(4) It is puzzling why the authors did not analyze end resection defects in rev7 deleted cells after a DSB. The author should employ the widely used resection assay after a HO break in rev3, rev7, and mre11 rev7 cells as described previously.

Thank you for the suggestion. Reviewer 1 also has raised this point. As suggested, we have analysed end resection in the rev7D cells at a HO inflicted DSB site using a qPCR assay (Mimitou and Symington, EMBO J. 2010; Gnugge et al., Mol. Cell 2023). The results revealed that deletion of REV7 led to an enhancement in the rate of DNA end resection at a DSB inflicted by HO endonuclease (Figure 9—figure supplement 3),

(5) Is it possible that Rev7 also contributes to NHEJ as the part of TLS polymerase complex? Although NHEJ largely depends on Pol4, the authors should not rule out that the observed NHEJ defect in rev7 cells is due at least partially to its TLS defect. In fact, both rev3 or rev1 cells are partially defective in NHEJ (Figure 7). Rev7-C1 is less deficient in NHEJ than REV7 deletion. These results predict that rev7-C1, rev3 should be as defective as the rev7 deletion. Additionally, the authors should examine if Rev7-C1 might be deficient in TLS. In this regard, does rev7-C1 reduce TLS and TLS-dependent mutagenesis? Is it dominant? The authors should also check if Rev3 or Rev1 are stable in Rev7 deleted or rev7-C1 cells by immunoblot assays.

We agree with the possibility that Rev7 may play a role in translesion DNA synthesis and TLS-dependent mutagenesis. Accordingly, Rev7-C1 might be deficient in TLS. While we do not rule out such scenarios, we respectfully suggest that this is outside the scope of the current manuscript. This manuscript focuses on the role of Rev7 in NHEJ and HR pathways, not on translesion DNA synthesis. Nevertheless, we recognise the importance of this line of investigation, and we will certainly consider this suggestion in our future work. Thank you.

(6) Due to the G4 DNA and G4 binding activity of REV7, it is not clear which class of events the authors are measuring in plasmid-chromosome recombination assay in Figure 9. Do they measure G4 instability or the integrity of recombination or both in rev7 deleted cells? Instead, the effect of rev7 deletion or rev7-C1 on recombination should be measured directly by more standard mitotic recombination assays like mating type switch or his3 repeat recombination.

We appreciate the Reviewer for highlighting this important point and would like to take the opportunity to clarify the rationale behind plasmid-chromosome recombination assay, as previously described (Paeschke et al., Cell 145, 678, 2011). In this assay, we are measuring the rate of Ura+ papillae formation arising from integration of the targeting plasmid into the genome at the ura3-1 locus of wild-type and rev7D cells. Analysis of PCR-generated DNA fragments indicate that pFAT10-G4 plasmid integrates at the ura3-1 genomic locus of rev7D cells, but not in the wild-type cells (Figure 9-figure supplement 2). Further, we also measured the stability of G4 DNA and the results indicate that it is stable in rev7D cells.

**Recommendations for the authors:**

**Reviewer 1 (Recommendations for the authors):**
(1) Title: The word 'choice' implies a regulator. Is that the model here? Alternatively, is it pathway properties that define the preference of usage?

This is an excellent suggestion. In the revised submission, we rephrased the title “*Saccharomyces cerevisiae* Rev7 promotes non-homologous end-joining by inhibiting Mre11 nuclease and Rad50 ATPase activities and Homologous recombination.”

(2) Line 83, Introduction: Titia De Lange proposed an alternative/complementary model for Shieldin and REV7 to support fill-in by DNA polymerases including Pol alpha. This should be discussed.

We thank the reviewer for pointing out that we have not discussed the work from Titia De Lange’s research group. We have now added new sentences to the Introduction to describe the alternative model involving Polα-primase fill-in synthesis (p3.2.7).

(3) Line 131: The paragraph title needs to change. 2-hybrid assays cannot establish direct interaction especially when analyzing yeast proteins by yeast 2-hybrid. I agree that direct interaction is established by other means later.

Per the Reviewer’s suggestion, we have deleted the word “directly” from the title of the paragraph.

(4) Figure 1 D-F: The purity of the Rev7-GFP fusion is shown in Figure S1, and the purity of the Rad50, Mre11, and Xrs2 subunits as assessed by PAGE should be shown as well.

Following this suggestion, we have included images of Coomassie blue-stained SDS-polyacrylamide gels (Figure 1-figure supplement 1), which show the purity and size of GFP tagged Rev7, Rad50, Mre11, Xrs2, Rev1, Sae2 and Dmc1 proteins.

(5) Please check the Kd values. In the graph in D, the differences between Rad50, Mre11, and Xrs2 look much larger than the values in F suggest.

This is a fair point and we appreciate the reviewer for highlighting. The differences between the binding profiles of the Rad50, Mre11, and Xrs2 with Rev7 as shown in the previous version of the manuscript were not obvious because of cluttering of binding curves. Therefore, the binding profiles of interacting pair of proteins were plotted separately to highlight the differences (Figure 1—figure supplement 3). Further, we rigorously analysed the dataset to ascertain the binding affinities and found that the *Kd* values obtained were in good agreement with the values shown in Figure 1D.

(6) Figure 1S3: Please label the bands.

In the revised manuscript, the protein bands in Figure1-figure (previously Figure 1S3) are identified with their names.

(7) Line 195: Change Figure 1 to Figure 1S4.

We have introduced the correction in the revised manuscript.

(8) Line 202: The minimal interaction domain of 42 aa is only described in the next paragraph. The description anticipates a result about the 42 aa fragment that has not been shown to this point. Please reorder results or descriptions to make this coherent.

We have implemented the change, as per the Reviewer’s suggestion.

(9) Figure 2: The two-hybrid analysis in Figures 1 and 2 also identifies Rev7 self-interaction, which is not discussed. This serves as another control against the artifact of the truncation proteins and should be discussed.

We have now discussed the significance of Rev7 self-interaction in the Y2H experiments wherever relevant in the text.

(10) Is the 42 aa fragment sufficient to elicit a two-hybrid signal?

We thank the reviewer for this insightful comment. To test this premise, we expressed the terminal 42 amino acid sequence of Rev7 using bait pGBKT7 vector. The results revealed that the 42 residue fragment of ScRev7 alone is sufficient for a two-hybrid interaction with the MRX subunits (Figure 2-figure supplement 1).

(11) Line 289: Why are the EMSA conditions described as physiological? As per Material and Methods, the reaction mixtures contain 20 mM Tris-HCl (pH 7.5), 0.1 mM DTT, 0.2 mg/ml BSA, and 5% glycerol, which is far from physiological.

As suggested by all three reviewers, the data showing the interaction of Rev7 and its truncation derivative Rev7-C1 with G4 DNA has been deleted in the revised version of the manuscript.

(12) Figure 4C: The figure needs to increase in size. The plotting symbols are not all visible, and it is undefined what the black squares represent.

Following the reviewer's suggestion, Figure 4C has been omitted in the revised version of the manuscript.

(13) Figure 5: The MRX nuclease assays were conducted in the presence of Manganese. Has the more physiological divalent cation magnesium been tested?

This has been addressed in response to the query of Reviewer 1 (Public Review). As noted above, Mre11 exhibits DNase activities only in the presence of Mn²⁺.

(14) In Figure 5D, lane 2: What is the concentration of Rev7?

We appreciate the reviewer for catching this. The concentration of ScRev7 used for the reaction shown in Figure 5D, lane 2 was 2 μM, as specified in the Figure legend.

(15) Figure 6 legend: Lane 1620 "same as in lane "Is there a "1" missing?

We thank the reviewer for pointing out the typographical error, which has been corrected in the revised manuscript.

(16) Figure 9: Rev7-C1 lacks the 42 a peptide that is postulated to mediate anti-resection but shows normal HR here. This seems unexpected based on the premise that the 42 aa fragment supports end-joining. Rev7 seems to suppress HR independent of the function of the 42 aa peptide.

This has been addressed in response to the query posed by Reviewer 1 in the Public Review. We do see that the Rev7-C1 lacking the 42 aa peptide suppresses HR, but the suppression was only partial as compared with the wild type. This is consistent with biochemical assays suggesting that Rev7-C1 exerts partial inhibition on the Mre11 nuclease (Figure 5) and Rad50 ATPase (Figure 6) activities. Further, the AF2 models indicate that, in addition to the C-terminal 42-aa region, other regions of Rev7 also interact with the Mre11 and Rad50 subunits (Figure 2—figure supplement 2), consistent with biochemical and genetic data.

(17) Line 478: The conclusion that "these findings are consistent with the idea that REV7 completely abolishes DSB-induced HR in *S. cerevisiae*." is overly broad as the assay

We agree with the reviewer's assessment. Accordingly, we have rephrased the sentence to soften the claim.

Line 483ff: Based on the comments on Figure 9, the introductory sentences of the discussion do not seem to be supported by the data, as Rev7 appears to regulate HR independent of the 42 aa peptide.

Please refer to the response of comment #16 above

(18) Line 536: Similarly to above 17, the conclusion about the effect of the 42 aa peptide on HR appears unwarranted.

We have revised the statement to moderate the previously exaggerated claims.

(19) In all figures, please list in the legend, which exact strains have been used referring to Table S5.

We have now included mentions of the strains in the figure legend wherever applicable.

(20) Line 351: linear.

It is corrected in the revised manuscript.

**Reviewer 2 (Recommendations For The Authors):**
(1) It is very strange and unusual that Rev7 independently binds to all three subunits of the MRX complex, raising a question of how specific these interactions are. At least, it should be a negative control in their YH2 assay and protein-protein interaction assay in vitro that Rev7 does not bind to some other proteins. For example, Sae2 and Rev7 interactions can be tested.

The reviewer is right that it is important to validate the specificity of Y2H interactions as well as in vitro enzyme assays. These findings are shown in Figure 6 and Figure 5-figure supplement 1. As suggested by the Reviewer, we included *SAE2* in Y2H and MST assays, and Dmc1 and Sae2 in vitro enzyme assays. Our results clearly showed that Sae2 neither interacts with MRX subunits in Y2H assays (Figure 1A-C) nor inhibits the Sae2’s nuclease and Dmc1’s ATPase activities in vitro (Figure 6 and Figure 5-figure supplement 1)

(2) It is surprising that in the Discussion the authors speculate that Rev7 might recruit Mus81 nuclease for cleavage, completely ignoring their own publication on the cleavage of G4 by MRX.

We agree with the reviewer, and we have added discussion about MRX (mentioned above by the reviewer) in revised version.

(3) How does the AlphaFold-Multimer modeling predict the interaction between Rev7 and MRX as a complex? Are the same regions of MRX accessible for the interaction with Rev7 in this case? Similarly, how are the activities of the MRX complex and phosphorylated Sae2 (see P. Cejka's work) affected by Rev7?

Thank you for pointing this out. In this study, we investigated the interaction between Rev7 and Mre11, and between Rev7 and Rad50 subunits using AF2 algorithm. However, the three-dimensional structure of *S. cerevisae* MRX-Rev7 complex could not be constructed due to the size limits imposed by AF2 algorithm. Therefore, we are unable to comment on whether the same regions of MRX subunits in the complex are accessible for the interaction with Rev7. That said, AF2 algorithm has recently been used for structural modelling of *S. cerevisiae* Mre11 (1–533)-Rad50 (1–260 + 1,057–1,312) complex (Nicolas *et al., Mol. Cell*
**84**, 2223, 2024). As such, there are no AF2 structural models that cover the whole length of Mre11-Rad50 proteins.

Regarding the second point raised by the Reviewer, our results suggest that Rev7 does interact with Sae2 in Y2H assays. However, whether phosphorylated Sae2 could potentially affect the interaction between MRX subunits and Rev7 warrants further studies.

Minor points:(1) Figure 1. The labeling of the strains in A and B is genes and in C is proteins.

The reviewer is correct. We have now corrected the error in the Figure 1 and 2.

(2) Abstract. Carefully check English grammar.

We thank the Reviewer for spotting this, which has been corrected in the revised manuscript.

(3) Line 322 "Further, it has been demonstrated that Mre11 cleaves non-B DNA structures such as DNA hairpins, cruciforms and intra- and inter-molecular G-quadruplex structures." It has not been shown that Mre11 cuts cruciform structures.

We thank the referee for spotting this error. Mre11 does not cleave cruciform DNA structures. This error is corrected in the revised manuscript.

(4) Page 14. Lines 452-455. What does "selective and non-selective media" mean? Is it without and with HU treatment?

Thanks very much for the comment. In our manuscript, selective medium is composed of SC/-Leu with HU and non-selective medium is without HU. We have clarified this point in the revised version.

(5) Page 15. Lane 472 "To assess whether increased frequency of HR is due to the instability of G-quadruplex DNA in rev7Δ cells, we examined the length of G4 DNA inserts in the plasmids carrying sequences during HR assay". It is not clear what does mean" during HR assay"? Did you examine the presence of G4 in Ura+ recombinants? If not, this analysis is meaningful.

The reviewer is correct. We measured the presence of G4 DNA insert in Ura+ recombinants. The text has been appropriately edited to reflect these necessary modifications.

(6) What is the nature of the ura3-1 allele? Can it revert to URA3 in rev7 mutants?

The *ura3-1* allele (glycine-to-glutamate substitution) reverts to Ura3+ at a low rate of ~2.5 × 10−9 in both orientations (Johnson et al., Mol. Cell 59, 163, 2015)

(7) From the way that the recombination process is depicted it seems that the authors believe that plasmid should integrate into the chromosome. In reality, in most cases it should be a gene conversion where the G4 sequence (if it indeed induces DSBs) should be replaced by the wild-type segment form ura3-1, integration is not required since it is 2-micron plasmid.

We apologize for not having made this clearer. The recombination assay with targeting plasmids containing G4 DNA forming sequences was performed as previously described (Paeschke et al., Cell 145, 678, 2011). In this assay, the appearance of Ura+ recombinants arise from the integration of the targeting plasmid bearing ura3G4 allele (with a G4 DNA forming insert) integrates into the genome at the ura3-1 locus. As shown in Author response image 1B, this is confirmed by PCR amplification of the insert in the genomic DNA of wild type and rev7D cells.

**Reviewer 3 (Recommendations For The Authors):**
(1) All Y2H experiments were performed with REV7 fusion to pGBKT7 and MRX to pGADT7. It will be helpful to test if pGAD-Rev7 also interacts with pGBK-Mre11 or Rad50 by Y2H.

Following the reviewers' suggestions, we performed Y2H experiments in wild-type PJ69-4a cells co-transformed with the pGBKT7 vector expressing MRX subunits and the pGADT7 vector expressing Rev7. The results indicated that Rev7 interacts with Mre11, Rad50 or Xrs2 subunits, indicating that interactions are vector-independent.

**Author response image 1. sa4fig1:** Yeast two hybrid analysis suggest interaction between Rev7 and MRX subunits. PJ69-4A cells were co-transformed with bait vector expressing Rev7 or the Mre11, Rad50 or Xrs2 subunits and prey vector expressing Rev7 protein. Equal number of cells were spotted onto –Trp – Leu and –Trp – Leu –His dropout plates containing 3-AT and images were obtained following 48 h of incubation at 30°C. The data is representative of three independent experiments.

(2) G4 studies are under-developed and do not add much or even negatively to the manuscript. The author might consider revising the manuscript to improve their integration with better rationales or logic. Alternatively, the authors should consider removing the G4 part for another paper.

This concern was also raised by Reviewer 1 and 2. Following the suggestions of all reviewers, figures and text related G4 DNA studies have been deleted in the revised manuscript.